# Don't Restart, Just Reuse: Reoptimizing MILPs with Dynamic Parameters

**Sijia Zhang** [1]  **Shuli Zeng** [1]  **Shaoang Li** [1]  **Feng Wu** [1]  **Shaojie Tang** [2]  **Xiang-Yang Li** [1]

## Abstract

Many real-world applications, such as logistics, routing, scheduling, and production planning, involve dynamic systems that require continuous updates to solutions for new Mixed Integer Linear Programming (MILP) problems. These systems often require rapid updates to their solutions to accommodate slight modifications in constraints or objectives introduced by evolving conditions. While reoptimization techniques have been explored for Linear Programming (LP) and certain specific MILP problems, their effectiveness in addressing general MILP is limited. In this work, we propose a two-stage reoptimization framework for efficiently identifying high-quality feasible solutions. Specifically, we first utilize the historical solving process information to predict a high confidence solution space for modified MILPs, which is likely to contain high-quality solutions. Building on the prediction results, we fix a part of variables within the predicted intervals and apply the Thompson Sampling algorithm to determine which variables to fix. This is done by updating the Beta distributions based on the solutions obtained from the solver. Extensive experiments across nine reoptimization datasets show that our VP-OR outperforms the state-of-the-art methods, achieving higher-quality solutions under strict time limits.

## 1. Introduction

Traditional combinatorial optimization problems require finding solutions for a single instance. However, many real-world scenarios, such as system control (Marcucci & Tedrake, 2020), railway scheduling (Zhang et al., 2020)

[1]School of Computer Science and Technology, University of Science and Technology of China [2]Department of Management Science and Systems, State University of New York at Buffalo. Correspondence to: Xiang-Yang Li <xiangyangli@ustc.edu.cn>, Feng Wu <wufeng02@ustc.edu.cn>.

*Proceedings of the 42$^{nd}$ International Conference on Machine Learning*, Vancouver, Canada. PMLR 267, 2025. Copyright 2025 by the author(s).

and production planning (Dunke & Nickel, 2023; Cedillo-Robles et al., 2020), involve systems that change dynamically over time. Thus, throughout the continuous operation of such systems, it is required to compute solutions for new Mixed Integer Linear Programming (MILP) problems, which are similar to the previous instances but differ in some parameters in specific model elements such as objective functions, constraints, and variable bounds. Traditionally, each of these new MILP instances is solved from scratch, which overlooks the opportunity to leverage valuable information from the previously solved instances. This approach can be computationally expensive. Additionally, it is challenging to generate high-quality operational plans within a short timeframe for time-critical applications.

Reoptimization techniques have been well-studied for the LP case (John & Yıldırım, 2008) and heuristic algorithms for some special MILP problems, e.g., the railway planning problem (Blair, 1998), general assignment problems (NAUSS, 1974) and other combinatorial problems (Libura, 1996; 1991; Sotskov et al., 1995). However, these techniques have limited applicability to general MILPs. The current reoptimization techniques for general MILP instances (Ralphs & Güzelsoy, 2006; Ralphs et al., 2010; Gamrath et al., 2015; Patel, 2024) can be divided into three categories: (1) reuse of historical optimal solutions (2) reuse of branch-and-bound trees (3) adjustment of parameters. The limitations of their approach are twofold. Firstly, the optimal solution from the original problem may no longer be valid for the new problem. This is because the range of variable values for the optimal solution can shift significantly, even with small modifications to the problem's parameters (Guzelsoy, 2009). Secondly, reusing branching strategies and adjusting parameters primarily saves time on decision-making within the branch-and-bound algorithm itself, such as selecting variables, generating heuristics, and constructing cutting planes. However, this approach does not reduce the overall size or complexity of the problem.

Recently, there has been an increased interest in end-to-end problem solving that generate high-quality solutions for MILPs (Nair et al., 2020; Han et al., 2023; Khalil et al., 2022; Ye et al., 2023; 2024). Previously, Neural Diving (ND) (Nair et al., 2020) is proposed to generate a Bernoulli distribution for the solution values of binary variables. They employ a selectivenet (Geifman & El-Yaniv, 2019) to learn

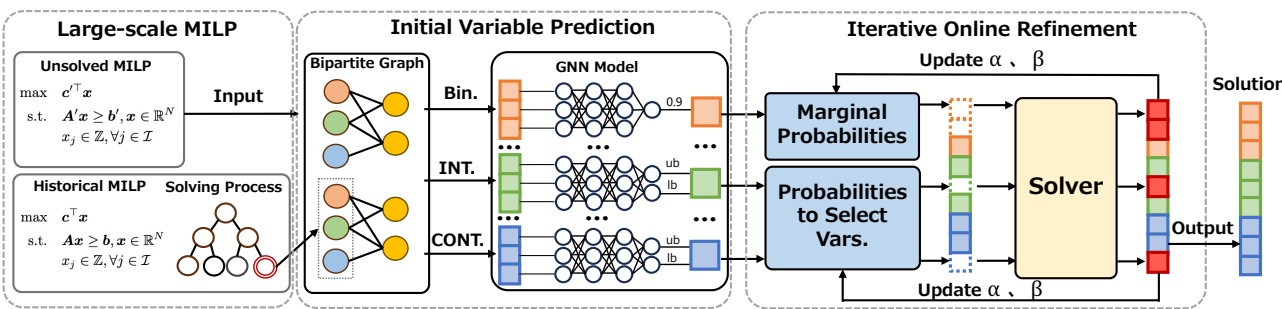

*Figure 1.* Illustration of our proposed two-stage reoptimization framework. Our approach first predicts a marginal probability of each binary variable and the feasible ranges of integer and continuous variables utilizing a graph neural network (GNN), and then employs the Thompson Sampling algorithm to iteratively select the variable to apply the interval.

which variables' predicted values to be fixed. The main disadvantage of ND is that fixing variables can lead to low-quality or infeasible solutions if the selectivenet fails to identify variables with inaccurate predictions. To mitigate the issue, current end-to-end methods (Han et al., 2023; Ye et al., 2024; Huang et al., 2024) employ a Predict-and-Search (PS) framework. They utilize large neighborhood search (LNS) (Carchrae & Beck, 2009) to explore solutions near the predicted values. These LNS-based approaches do not differentiate between variables predicted with greater accuracy. Instead, they predetermine the number of variables to search around (controlled by hyperparameters) and uniformly apply uncertainty neighborhoods around these binary variables. These methods are not directly suited for reoptimization scenarios, which urgently require quickly obtaining high-quality feasible solutions (Marcucci & Tedrake, 2020; Zhang et al., 2020). For example, instances in the 'bnd_3' dataset from the reoptimization competition (Bolusani et al., 2023) cannot find a primal solution within 300 seconds using these methods.

In this paper, we primarily focus on finding primal solutions for MILP problems with dynamic parameters. We propose a two-stage reoptimization framework, which consists of a **V**ariable **P**rediction model and an **O**nline **R**efinement module (VP-OR). We find that over 26.6% integer and continuous variables cannot be accurately predicted if relying solely on optimal solution values, as shown in Table 10. To overcome this, we propose a method to predict the bounds for these variables, leveraging historical branch-and-bound processes from previously solved instances. Inspired by the recent success for variable prediction (Han et al., 2023; Huang et al., 2024), VP-OR employs graph neural networks (GNNs) (Gori et al., 2005) to analyze changes in problem structure and significantly improves the accuracy of variable predictions. To meet time-critical demands, the online refinement module adopt a fixing strategy similar to ND (Nair et al., 2020), achieving solving times at least 10 times shorter

per iteration compared to traditional LNS strategies, as shown in Table 2. Different from ND and PS (Han et al., 2023), we utilize Thompson Sampling (Thompson, 1997) to differentiate between variables predicted with greater accuracy. We fix a subset of variables to apply the prediction intervals and iteratively improve the sampling based on solution results from each round.

Empirically, we compare VP-OR against the leading reoptimization method (Patel, 2024), two end-to-end machine learning-based baselines (Nair et al., 2020; Han et al., 2023), and the open-source solver SCIP (Bestuzheva et al., 2021) across nine reoptimization datasets. The results indicate that VP-OR outperforms the other methods in delivering highly accurate solutions under strict time limits. In addition, we evaluate the performance over a longer duration, revealing that VP-OR converges more rapidly, achieving smaller primal gaps compared to the other methods.

## 2. Related work

**Reoptimization for MILPs.** Early methods (Ralphs & Güzelsoy, 2006; Ralphs et al., 2010) were primarily based on duality theory and focused on sequences of MILPs where only the right-hand side changes. These approaches leveraged dual information obtained through primal algorithms to enable "warm starting", accelerating the resolution of subsequent problems. Subsequent research (Gamrath et al., 2015) extended these methods to handle broader scenarios, incorporating techniques such as reusing branch-and-bound trees. In these approaches, the modified problem is treated as a subproblem of the base problem, or, if only the objective function changes, the search can "continue" from the last known search boundary. Specifically, these methods use the leaf nodes of the base problem's branch-and-bound tree as starting points for solving the modified problem. Building on this, more recent work (Patel, 2024) addressed even more complex reoptimization scenarios, where, apart from the

objective function but also other parameters-such as variable bounds, matrix coefficients, and constraint right-hand side values-can undergo upward or downward perturbations. In the approach, a series of past solutions is preserved, allowing the method to reuse these solutions for the new problem. They also explore the reuse of branching strategies and the adjustment of parameters related to the cutting planes and heuristic algorithms, fine-tuning the solver's behavior to better tackle the modified problem. They primarily relied on heuristic adjustments, lacking the ability to predict the changes in optimal solutions.

**Prediction for integer variables.** Many real-world scenarios that rely on reoptimization techniques naturally involve both integer and continuous variables. For example, production quantities in manufacturing are integers (Cedillo-Robles et al., 2020), while power levels in energy optimization problems are continuous variables (Yokoyama et al., 2002). Despite this, most existing end-to-end machine learning-based methods (Han et al., 2023; Khalil et al., 2022; Huang et al., 2024) primarily focus on predicting solutions for binary variables. Neural Diving (Nair et al., 2020) (ND) first proposed to represent integer variables in binary form and predict each bit's value. To address the issue of representing integer variables with excessively many bits, ND considered the first few bits as the most significant during the solving process and introduced a hyperparameter to control the maximum number of bits predicted for each variable. Building on ND, Ye et al. (2024) introduced a generalized confidence threshold method. However, predicting each binary bit's actual value can lead to inaccuracies, causing overall prediction errors.

## 3. Preliminaries

**Mixed integer linear programming(MILP).** Given a set of decision variables $x \in \mathbb{R}^n$, the MILP problem is formulated as follows:

$$
\begin{aligned}
\min \quad & c^\top x, \\
\text{s.t.} \quad & Ax \geq b, \quad l \leq x \leq u, \\
& x \in \{0,1\}^p \times \mathbb{Z}^q \times \mathbb{R}^{n-p-q},
\end{aligned}
\tag{1}
$$

where $c \in \mathbb{R}^n$ is the objective coefficients, $A \in \mathbb{R}^{m \times n}$ is the constraint coefficient matrix, $b \in \mathbb{R}^m$ is the right-hand side vector, $l, u \in \mathbb{R}^n$ are the variable bounds.

**Modified MILP problem.** We consider scenarios similar to the previous reoptimization work (Patel, 2024), involving a series of MILP instances based on an MILP (historical instance) taken from a specific application. Each subsequent instance (modified instance) is modified from the previous one with random perturbations and rotations to parameters such as the objective vector, constraints, and variable bounds. The previous instances has been solved to

optimality. They provide not only the optimal solution but also detailed records of intermediate computational steps, such as selected branches and basis variables at each node's LP relaxation. These records can be strategically leveraged in the reoptimization algorithm to accelerate the solving process for the modified instances.

**Bipartite graph for MILPs** An MILP problem can be effectively represented as a weighted bipartite graph $G = (V \cup C, E)$ (Nair et al., 2020; Gasse et al., 2019). Each vertex in $V$ corresponds to a variable of the MILP, and each vertex in $C$ represents a constraint. An edge $(v_i, c_j)$ connects a variable vertex $v_i$ with a constraint vertex $c_j$ if the variable is involved in the constraint. The edge set $E \in \mathbb{R}^{m \times n \times e}$ represents the edge features, where $m$ and $n$ denote the number of constraints and variables, respectively, and $e$ indicates the dimension of the edge attributes.

**Online contextual thompson sampling.** Thompson Sampling is a heuristic strategy used in decision-making scenarios like the multi-armed bandit (MAB) problem (Agrawal & Goyal, 2012; Zhao, 2022). This method is used for choosing actions according to their expected rewards, which are continuously updated using Beta probability distributions $Beta(\alpha, \beta)$ (Gupta & Nadarajah, 2004). The Beta distribution forms a family of continuous probability distributions over the interval $(0, 1)$. The probability density function (pdf) of a $Beta(\alpha, \beta)$ distribution, where $\alpha > 0$ and $\beta > 0$, is given by: $f(x; \alpha, \beta) = \frac{\Gamma(\alpha+\beta)}{\Gamma(\alpha)\Gamma(\beta)} x^{\alpha-1}(1-x)^{\beta-1}$, where $\Gamma(\cdot)$ is the Gamma function. The mean of the $Beta(\alpha, \beta)$ distribution is $\frac{\alpha}{\alpha+\beta}$, and as the parameters $\alpha$ and $\beta$ increase, the distribution becomes more concentrated around the mean. The beta distribution is useful for Bernoulli rewards because if the prior is a $Beta(\alpha, \beta)$ distribution, then after observing a Bernoulli trial, the posterior distribution is simply $Beta(\alpha + 1, \beta)$ or $Beta(\alpha, \beta + 1)$, depending on whether the trial resulted in a success or failure, respectively.

## 4. Methodology

In this section, we present the insights and details of our VP-OR framework. We start by utilizing the historical information to predict a feasible interval containing the modified problem's optimal solution in Section 4.1. Then, we employ the Thompson Sampling algorithm to refine the solving space in Section 4.2. Figure 1 shows the overall procedure of our VP-OR framework.

### 4.1. Initial variable prediction

#### 4.1.1. GRAPH REPRESENTATION

The feature extraction process is divided into two parts: the historical instance and the modified instance. For the modi-

*Table 1.* Comparison of variable prediction accuracy for different datasets. This table presents the number of variables and mispredicted variables across different types (binary, integer, and continuous) when using GNN-based predictions. Mispredicted variables represent those whose predicted bounds or values differ from the optimal solution.

| Var. num. | bnd_1 | bnd_2 | bnd_3 | mat_1 | obj_1 | obj_2 | rhs_1 | rhs_2 | rhs_3 |
|---|---|---|---|---|---|---|---|---|---|
| binary var. | 2993.0 | 1457.0 | 1457.0 | 500.0 | 360.0 | 355.0 | 12510.0 | 500.0 | 500.0 |
| mispredicted binary var. | 8.2 | 6.7 | 4.5 | 37.4 | 5.6 | 0.2 | 64.3 | 0.0 | 0.0 |
| integer var. | 124.0 | 0.0 | 0.0 | 0.0 | 0.0 | 150.0 | 0.0 | 0.0 | 0.0 |
| mispredicted integer var. | 17.4 | 0.0 | 0.0 | 0.0 | 0.0 | 5.0 | 0.0 | 0.0 | 0.0 |
| continuous var. | 0.0 | 301.0 | 301.0 | 302 | 0.0 | 240.0 | 250.0 | 500.0 | 500.0 |
| mispredicted continuous var. | 0.0 | 0.0 | 2.0 | 0.0 | 0.0 | 16.8 | 0.0 | 1.6 | 4.7 |

fied instance, we represent it using a classic bipartite graph structure (Gasse et al., 2019). For the historical instance, we aim to extract additional historical solving information to predict how the optimal solution may change under small perturbations in the MILP.

In MILP, integer variables are often relaxed to continuous values to apply duality concepts. However, the dual problem from the relaxed problem may not directly reflect the relationship between the optimal solution and constraints under integer restrictions. We address this challenge by leveraging a key property of branch-and-bound trees: the final leaf node that yields the MILP optimal solution has the characteristic that its LP relaxation solution is also an integer solution. The leaf node represents a subproblem of the original MILP, distinguished by the addition of a series of branching constraints. We include the feasible basic variables and dual solutions of the leaf node as features, which are commonly applied in LP sensitivity analysis (Higle & Wallace, 2003), aiming to capture which variables and constraints are sensitive to parameter changes. This approach significantly improves the accuracy of binary variable predictions compared to traditional end-to-end solving methods, which rely solely on modeling the problem as a bipartite graph and optimal solution values. In Appendix E.5, we present the comparison results between the reuse of historical solving information and the traditional vanilla bipartite graph predictions. The features used in the graph representation is detailed in Table 5 in Appendix A.

### 4.1.2. GNN-BASED INITIAL VARIABLE PREDICTION

Classic end-to-end approaches (Khalil et al., 2022; Han et al., 2023) are specifically designed for binary variables and predict a n-dimension vector $(p_\theta(x_1 = 1; M), \ldots, p_\theta(x_p = 1; M))$ to represent the conditional probability of $p$ binary variables. However, these methods can not work well in many real-world scenarios, which mainly contain integer and continuous variables. For instance, in the dataset named "vary_matrix_rhs_bounds" in the MIP Workshop 2023 Computational Competition (Bo-

lusani et al., 2023), there are 27,710 variables but only 400 binary variables.

**Loss function.** We use a categorical cross-entropy loss function to train the GNN. For binary variables, the model predicts probabilities that are directly compared with the actual binary values to compute the optimal solution. In the datasets, maximum integer values exceed 100,000, which would require at least 18 bits for binary representation. Using a direct approach without these transformations results in out-of-memory errors due to large output dimensions. Therefore, for integer and continuous variables, they are first represented as 8-bit binary numbers using the method described in Appendix B, which involves using the binary representation of the logarithmic value. The use of binary representation and logarithmic transformation addresses practical computational constraints. These binary representations are then used as labels and compared with the predicted probabilities.

**Prediction confidence processing.** We apply the confidence threshold method (Yoon, 2022) to filter the predicted probabilities and distinguish between confident and uncertain predictions. For the binary digits with high confidence, the binary digits are fixed to their predicted values. For uncertain binary digits (i.e., those with probabilities between 0.1 and 0.9), we allow them to vary between 0 and 1. We establish the upper and lower bounds of the predicted binary encoding by setting the uncertain binary digit to its maximum value 1 for the upper bound and its minimum value 0 for the lower bound.

### 4.2. Iterative online refinement

#### 4.2.1. OBSERVATION

We test the prediction accuracy of GNN-based models on a variety of datasets, which were carefully selected to represent different types of parameters, including variable bounds, objective function coefficients, matrix parameters, and right-hand side constraints. Table 1 provides the number of variables and the mispredicted variables for each dataset. The

*Table 2.* Comparison of the solving time (second) using variable fixing and large neighborhood search (LNS) methods, with different percentages of variables (50% and 70%). For LNS methods, feasible solutions are searched around the predicted values for these variables. For variable fixing methods, these variables are set to the predicted values.

| Solving time | bnd_s1 | bnd_s2 | bnd_s3 | mat_s1 | obj_s1 | obj_s2 | rhs_s1 | rhs_s2 | rhs_s3 |
|---|---|---|---|---|---|---|---|---|---|
| SCIP original solving time | 356.09 | 314.35 | 574.72 | 541.78 | 570.69 | 200.10 | 546.40 | 68.67 | 90.24 |
| LNS (50% binary variables) | 328.50 | 146.38 | 427.19 | 111.54 | 123.71 | 306.21 | 287.91 | 59.85 | 77.08 |
| LNS (70% binary variables) | 335.51 | 265.82 | 307.43 | 497.24 | 703.86 | 307.78 | 247.45 | 81.27 | 80.08 |
| Fix 50% variables (only binary) | 17.61 | 13.44 | 77.83 | 0.60 | 65.61 | 5.71 | 9.99 | 29.00 | 25.13 |
| Fix 70% variables (only binary) | 4.92 | 6.46 | 32.94 | 0.38 | 4.22 | 3.08 | 5.42 | 13.78 | 12.41 |
| Fix 50% variables (all) | 3.87 | 2.76 | 7.42 | 0.57 | 49.65 | 0.24 | 2.71 | 3.97 | 8.73 |
| Fix 70% variables (all) | **0.71** | **2.44** | **4.02** | **0.37** | **1.04** | **0.23** | **0.47** | **3.53** | **4.18** |

results indicate that inaccuracies in predicted values are typically concentrated in a small subset of the variables. This is especially evident with integer and continuous variables, where less than 14% of the ranges deviate from the true feasible regions.

With this observation, it is reasonable to accelerate the solving process for MILP problems by fixing variables in the partial solution. To simplify the formulation, we denote the constraint space of the modified instance as: $S = \{\boldsymbol{x} \in \{0,1\}^p \times \mathbb{Z}^q \times \mathbb{R}^{n-p-q} : (\boldsymbol{A} + \Delta\boldsymbol{A})\boldsymbol{x} \geq (\boldsymbol{b} + \Delta\boldsymbol{b}), (\boldsymbol{l} + \Delta\boldsymbol{l}) \leq \boldsymbol{x} \leq (\boldsymbol{u} + \Delta\boldsymbol{u})\}$. Specifically, the sub-problem of an instance using the fixing strategy with the predicted binary value $\tilde{\boldsymbol{x}}$, the predicted lower bound $\tilde{\boldsymbol{l}}$ and upper bound $\tilde{\boldsymbol{u}}$ can be formulated as:

$$\min_{x \in S(\tilde{x}, I) \cap S} \quad (\boldsymbol{c} + \Delta\boldsymbol{c})^\top \boldsymbol{x} \qquad (2)$$

where the learning-based constraint set $S(\tilde{x}, I)$ is defined as: $S(\tilde{x}, I) = \{x \in \{0,1\}^p \times \mathcal{Z}^q \times \mathcal{R}^{n-p-q} : x_i = \tilde{x}_i, i \in \{1, 2, \ldots, p\} \cap I, \tilde{l}_j <= x_j <= \tilde{u}_j, j \in \{p+1, p+2, \ldots, n\} \cap I\}$, and $\tilde{x}_i$ represents the predicted probability for binary variables $x_i$, and $\tilde{l}_j$ and $\tilde{u}_j$ are the predicted lower and upper bounds for integer or continuous variables $x_j$. Here, $I$ is a subset of $\{1, 2, \ldots, n\}$, representing the set of selected related variables in the constraint set. However, for $i \in I$, if $\tilde{x}_i \neq x_i^*$ or if $\tilde{l}_j > x_j^*$ or $\tilde{u}_j < x_j^*$, where $x_i^*$ denotes the optimal value of the variable $x_i$ in the modified problem, the fixing strategy may lead to suboptimal solutions or even infeasible sub-problems. Identifying the appropriate set $I$ to avoid these inaccurate predictions is challenging, particularly when handling large-scale problems where the search space is vast, and the number of variables is substantial.

Interestingly, through a number of experimental tests, as shown in the Table 2, we found that when fixing a portion of the variables, the solution time of the problem can become very short. For comparison, we included the solving times of the default SCIP and those obtained by large neighborhood

search (LNS) methods. In cases where incorrect variable fixing caused infeasibility, we randomly select variables multiple times and compute the average time taken to find feasible solutions across these selections. The results show that we can increase the problem solving efficiency to 3-10 times by fixing only the binary variables when we add the estimated integer and continuous variables. It motivates us to choose variables based on the feedback of each solution, and gradually update the initial values of predictions based on the solution values of the binary variables.

### 4.2.2. ONLINE VARIABLE FIXING STRATEGY

Our approach is based on a simplifying assumption commonly used in prior work (Han et al., 2023) that treats each variable as independent of others. This assumption enables us to update the values of $\alpha$ and $\beta$ separately for each variable. During the initial presolve phase, we follow the approach of previous reoptimization work (Patel, 2024) by providing solutions from previous instances as hints to support the "completesol" heuristic method, effectively leveraging the historical solution as a warm start.

**Problem statement.** We model our problem as a stochastic multi-armed bandit (MAB) problem (Zhao, 2022). In the context of the solving process of MILP discussed, which variables to be apply the predicted values is analogous to an arm in a contextual MAB setup. At each round $t$, the corresponding variable set are fixed to their predicted values, which reduces the original problem to a subproblem. We then solve this subproblem and obtain a reward $r_{a_t} \in \{0, 1\}$. If the solution obtained in this iteration improves upon all previous solutions, we set $r_{a_t} = 1$; otherwise, $r_{a_t} = 0$. The objective is to discover better solutions with as few iterations as possible, approaching the optimal solution efficiently. This aligns with the typical MAB goal of maximizing the expected total reward over a time horizon T, i.e., $\mathbb{E}\left[\sum_{t=1}^{T} r_{a_t}\right]$, where $a_t$ represents the arm played at time $t$, and the expectation is taken over the random choices of

*Table 3.* Number of solved problems within 10s time limit for each method across datasets.

| Methods | bnd_1 | bnd_2 | bnd_3 | obj_1 | obj_2 | mat_1 | rhs_1 | rhs_2 | rhs_3 |
|---|---|---|---|---|---|---|---|---|---|
| SCIP | 5/5 | 0/5 | 0/5 | 5/5 | 5/5 | 5/5 | 5/5 | 5/5 | 5/5 |
| ND | 0/5 | 0/5 | 0/5 | 5/5 | 5/5 | 0/5 | 0/5 | 0/5 | 0/5 |
| PS | 5/5 | 0/5 | 0/5 | 5/5 | 5/5 | 5/5 | 5/5 | 5/5 | 5/5 |
| Re_Tuning | 5/5 | 3/5 | 3/5 | 5/5 | 5/5 | 5/5 | 5/5 | 5/5 | 5/5 |
| VP-OR(Ours) | 5/5 | 5/5 | 5/5 | 5/5 | 5/5 | 5/5 | 5/5 | 5/5 | 5/5 |

$a_t$ made by the algorithm.

We do not directly use the solution's objective value as the reward, as the reward may cause the model to favor solutions that are "good but not optimal", reducing the motivation for exploration. By rewarding only when the current solution is better than all previous ones, we can more clearly distinguish which arms lead to true improvements.

We employ an iterative thompson sampling approach to refine solutions. Following the results from each iteration, we update the beta distributions of variables for the subsequent round of optimization. For binary variables, the prior distribution is initialized as $\text{Beta}(p_i + 10^{-5}, 1 - p_i + 10^{-5})$, where $p_i$ represents the marginal predicted probability of the binary variable being fixed to 1. For integer and continuous variables, the prior is set as $\text{Beta}(1, 1)$. The value $\mu$, sampled from the Beta distribution, represents the probability of achieving a better solution (i.e., obtaining $r = 1$) by fixing binary variables to 1 or by imposing predicted bounds on integer variables. We select the top $a\%$ of these variables based on their $\mu_j$ values. The priors are updated according to the solving values for the unselected variables and based on whether the current solution improves for the selected variables. In our algorithm, we aim to avoid being trapped in local optima. To encourage exploration, no penalty is given when the current solution performs worse than previous iterations. Details of our thompson sampling algorithm are presented in Appendix C.

We propose the relaxation mechanism to address infeasible instances, which is detailed in Appendix D. It divides the fixed variables into ten groups and subsequently solves each without these variable sets. When a feasible solution cannot be found, we repeatedly apply the relaxation mechanism, building upon previous relaxations. Each iteration of this mechanism reduces the number of fixed variables. With enough iterations, the variables causing conflicts with the constraints are filtered out.

# 5. Experiments

Our experiments consist of three main parts: **Experiment 1**: Evaluate different methods on nine public reoptimization datasets, focusing on whether they can quickly find feasible solutions within the 10-second time limit. **Experiment 2**: Assess the quality of the feasible solutions obtained within the 10-second limit. **Experiment 3**: To provide a more intuitive comparison of solution convergence speeds, we plot the relative primal gap over time under a larger time limit of 100 seconds.

## 5.1. Experimental setup

**Benchmarks.** We select 9 series of instances from the MIP Computational Competition 2023 (Bolusani et al., 2023) to evaluate our approach. Each series has 50 similar instances with one or more components changing across instances. These instances need SCIP to solve from 60 to 600 seconds. Depending on the series, one of the following input can vary: (1) objective function coefficients (*obj_1, obj_2*), (2) variable bounds (*bnd_1, bnd_2, bnd_3*), (3) constraint right-hand sides (*rhs_1, rhs_2, rhs_3*), (4) constraint coefficients (*mat_1*). Most of these series are based on instances from the MIPLIB 2017 benchmark library (Gleixner et al., 2021), which is a well-known and widely used collection of benchmark problems in the field of MILP. They are frequently maintained and updated to include a diverse set of test problems sourced from various real-world applications and industries. Some of other series are collected from the real-world industrial use case and traditional problems. They are highly relevant to real-world reoptimization scenarios. Due to limited space, please see Appendix E.1 for details of these datasets.

**Training.** Each dataset contains 50 instances. To facilitate the experiments, we pair the instances in groups of two, resulting in 25 groups, including 20 groups in the training set and 5 groups in the test set. The first instance in each group serves as the historical instance, for which intermediate solving information required for feature extraction is pre-recorded. The specific features are detailed in the Appendix A. All numerical results are reported for the test set. To further increase the number of test samples, we generate similar datasets using *bnd_1* as a testing example, employing a method consistent with that published by the competition organizers. The results presented in Appendix E.8 are consistent with the tests shown in Table 4 of the main text. The model was implemented in PyTorch (Paszke et al.,

*Table 4.* Policy evaluation on the datasets, where "-" represents cases where the method could not find a feasible solution. The best performance is marked in bold.

| Methods | bnd_1 | | | bnd_2 | | | bnd_3 | | |
|---|---|---|---|---|---|---|---|---|---|
| | gap_abs | gap_rel | wins | gap_abs | gap_rel | wins | gap_abs | gap_rel | wins |
| SCIP | 1974.20 | 0.16 | 1/5 | - | - | - | - | - | - |
| ND | - | - | - | - | - | - | - | - | - |
| PS | 9665.20 | 0.81 | 0/5 | - | - | - | - | - | - |
| Re_Tuning | 1425.5 | 0.12 | 0/5 | - | - | - | - | - | - |
| VP-OR(Ours) | **299.40** | **0.02** | **4/5** | **40.20** | **0.11** | **5/5** | **28.60** | **0.06** | **5/5** |

| Methods | mat_1 | | | obj_1 | | | obj_2 | | |
|---|---|---|---|---|---|---|---|---|---|
| | gap_abs | gap_rel | wins | gap_abs | gap_rel | wins | gap_abs | gap_rel | wins |
| SCIP | 14.10 | 0.23 | 0/5 | 11.40 | 0.00 | 0/5 | 626.52 | 0.39 | 0/5 |
| ND | - | - | - | 11.40 | 0.00 | 0/5 | 674.21 | 0.44 | 0/5 |
| PS | 14.10 | 0.23 | 0/5 | 13.40 | 0.00 | 0/5 | 266.10 | 0.21 | 0/5 |
| Re_Tuning | 30.06 | 0.48 | 0/5 | 10.25 | 0.00 | 0/5 | **74.10** | 0.09 | 0/5 |
| VP-OR(Ours) | **10.09** | **0.16** | **5/5** | **3.28** | **0.00** | **5/5** | 0.64 | **0.00** | **5/5** |

| Methods | rhs_1 | | | rhs_2 | | | rhs_3 | | |
|---|---|---|---|---|---|---|---|---|---|
| | gap_abs | gap_rel | wins | gap_abs | gap_rel | wins | gap_abs | gap_rel | wins |
| SCIP | 173.08 | 0.50 | 0/5 | 12.29 | 0.00 | 0/5 | 15.01 | 0.00 | 0/5 |
| ND | - | - | - | - | - | - | - | - | - |
| PS | 67090.50 | 193.04 | 0/5 | 22.25 | 0.00 | 0/5 | 18.00 | 0.00 | 0/5 |
| Re_Tuning | 6.40 | 0.02 | 0/5 | 2.24 | 0.00 | 0/5 | 0.40 | 0.00 | 0/5 |
| VP-OR(Ours) | **0.73** | **0.00** | **5/5** | **1.85** | **0.00** | **5/5** | **0.26** | **0.00** | **5/5** |

2019) and optimized using Adam (Kingma & Ba, 2014) with training batch size of 16. The training process is conducted on a single machine that contains eight GPU devices (NVIDIA GeForce RTX 4090) and two AMD EPYC 7763 CPUs. Each instance uses only one GPU for both training and inference.

**Baselines.** We compare our approach against four baselines: the state-of-the-art open-source solver *SCIP* (Bestuzheva et al., 2021), the leading reoptimization method *Re_Tuning* (Patel, 2024), which won first place at the MIP Workshop 2023 competition (Bolusani et al., 2023) and does not rely on machine learning, and two GNN-based machine learning methods. In this paper, we compare *ND* (Nair et al., 2020) and *PS* (Han et al., 2023) as representative end-to-end approaches, both of which have gained significant popularity in recent years. *PS* is primarily based on the large neighborhood search (LNS) method, while *ND* utilizes a variable-fixing strategy for optimization. Please see Appendix E.2 for implementation details of these baselines. To provide a more comprehensive comparison with our approach, we also provide results for SCIP using the historical solution as a warm-start strategy in Appendix E.7. Due to the space limit, we

generate large-scale datasets IS and CA using the code from Gasse et al. (Gasse et al., 2019) and provide the results in Appendix E.10.

**Evaluation metrics.** For each instance, we first solve the problem without a time limit and record the optimal solution's objective value as $OPT$. Then, we apply a time limit of 10 seconds for each method. The best objective value obtained within the time limit is denoted as $OBJ$. We define the following performance metrics: **(1) Solve number**: This is the most fundamental metric, tracking the number of times a method successfully finds a feasible integer solution within the 10-second time limit. **(2) Gap**: We define the absolute and relative primal gaps as: $gap\_abs = |OBJ - OPT|$ and $gap\_rel = |OBJ - OPT|/(|OPT| + 10^{-10})$, respectively, and use them as performance metrics. Clearly, a smaller primal gap indicates a stronger performance. **(3) Wins**: This metric counts the number of instances where each method achieved the closest solution to the optimal one within the same time limit, relative to the total number of instances.

Throughout all experiments, we use SCIP 8.0.4 (Bestuzheva et al., 2021) as the backend solver, which is widely used in research of machine learning for combinatorial optimiza-

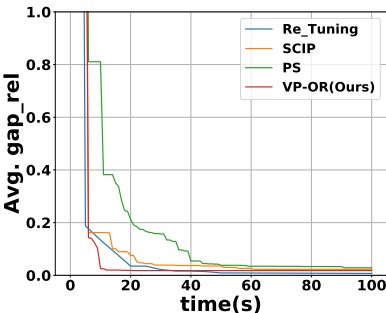 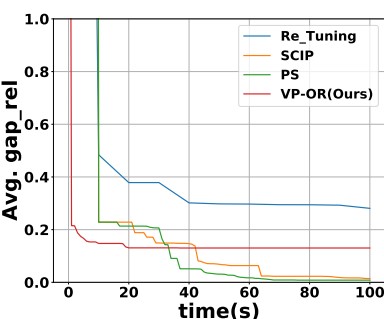 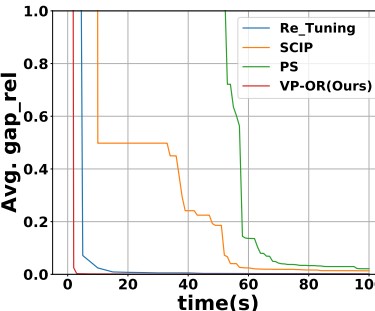

*Figure 2.* Performance comparisons in bnd_1, mat_1 and rhs_1, where the y-axis is average relative primal gap; each plot represents one benchmark dataset.

tion (Chmiela et al., 2021; Khalil et al., 2022; Gasse et al., 2019). We keep all the other SCIP parameters to default and emphasize that all of the SCIP solver's advanced features, such as presolve and heuristics, are open.

### 5.2. Experimental results

In our experiments, we include only one parameter: the percentage of fixed variables $P$. In this section, we present the results for $P = 0.7$. Results for other values of $P$ are provided in Appendix E.3.

**Experiment 1.** The results in Table 3 show how each method performs under the 10-second time constraint to find a feasible solution, which reflects the real-world need for quickly obtaining high-quality reoptimization solutions (Marcucci & Tedrake, 2020; Zhang et al., 2020). We observe that only our method, VP-OR successfully found feasible solutions across all datasets within the time limit. The reoptimization method, Re_Tuning, also performed relatively well compared to other methods. This improved performance can be attributed to its use of warm-starting with solutions from previous instances and parameter tuning using historical solving information.

**Experiment 2.** We evaluate the quality of the best feasible solutions found by different methods within the 10-second time limit. The evaluation is conducted across various datasets, with performance measured by absolute gap (*gap_abs*), relative gap (*gap_rel*), and the number of wins (*wins*), where wins indicate the number of instances for which a method achieves the best solution. The results are shown in Table 4, where "-" represents cases where the method could not find a feasible solution. In terms of *wins* and *gap_rel*, VP-OR surpasses all baseline methods. VP-OR performs exceptionally well in scenarios involving changes to variable bounds, matrix coefficients, and constraint right-hand sides. Specifically, in datasets where variable bounds are altered (e.g., *bnd_2*, and *bnd_3*), VP-OR achieves the

average relative gap close to 0.1 in 10 seconds, while other methods struggle to provide feasible solutions within 100 seconds. Additionally, Re_Tuning outperforms both SCIP and end-to-end prediction-based methods on most datasets. ND and PS might be more suitable for problems that are not time-sensitive and allow for longer solving times.

**Experiment 3.** To provide a more intuitive comparison of solution convergence speeds, we plot the relative primal gap over time with a larger time limit of 100 seconds in Figure 2, highlighting how our approach converges compared to other methods. We observe that VP-OR is more suitable for scenarios that require rapidly obtaining high-quality solutions in the short term. It converges quickly to find high-quality feasible solutions in the early stages of solving, but in the global scope, we also found that our method may encounter the possibility of getting stuck at suboptimal solutions. While Re_Tuning and LNS also show potential, it's noteworthy that in certain cases, SCIP performs even better than some of the optimization methods. Due to space constraints, we only present the results from three datasets in this section, with additional results provided in Appendix E.4.

## 6. Conclusion

This paper proposes VP-OR, a two-stage reoptimization framework for MILPs with dynamic parameters. VP-OR first trains a GNN model to predict the marginal probability of each binary variable and the feasible ranges of integer and continuous variables in the modified MILP instance. Further, the Thompson Sampling algorithm is employed to iteratively select which variables to apply the predicted intervals. Experimental evaluations demonstrate that VP-OR significantly improves the solution quality in the reoptimization setting under very strict time limits over the baselines. We anticipate and encourage further efforts to extend the applicability of VP-OR in ultra-large-scale scenarios.

## Acknowledgements

The research is partially supported by China National Natural Science Foundation with No. 62132018 , "Pioneer" and "Leading Goose" R&D Program of Zhejiang", 2023C01029, the University Synergy Innovation Program of Anhui Province with No. GXXT-2019-024, and the USTC Kunpeng-Ascend Scientific and Educational Innovation Excellence Center.

## Impact Statement

This paper presents work whose goal is to advance the field of machine learning for combinatorial optimization. There are many potential societal consequences of our work, none which we feel must be specifically highlighted here.

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

## A. More details of graph features.

The feature extraction process is split into two parts: the historical instance and modified instance. For the historical instance, we extract a richer set of graph features, including intermediate solving information. For the modified instance, we focus on structural information as Gasse et al. (2019). A list of the features used in our graph representation of the historical instance is detailed in Table 5.

*Table 5.* Description of the constraint, variable and edge features in our graph representation of the historical instance.

| Category | Feature | Description |
|---|---|---|
| variable vertex | lb | Original lower bound. |
| | ub | Original upper bound. |
| | objective_coeff | Objective coefficient. |
| | var_type | Type (binary, integer and continuous) as a one-hot encoding. |
| | leaf_lb | Lower bound of the leaf node which contains the optimal solution. |
| | leaf_ub | Upper bound of the leaf node which contains the optimal solution. |
| | depth | Depth of the leaf node. |
| | estimate | Estimate value of the leaf node. |
| | isBasic_var | If the variable is a basic variable in the LP relax of the leaf node. |
| | optvalue | Variable value in the optimal solution. |
| constraint edge | coef | Constraint coefficient. |
| constraint vertex | rhs | Right-hand side of the constraint. |
| | cons_type | Constraint type feature (eq, geq) as a one-hot encoding. |
| | isBasic_cons | If the constraint is a basic vector in the LP relax of the leaf node. |

## B. More details of logarithmic transformation

To reduce the dimensionality of integer variables, we apply a logarithmic transformation before converting the integer values into binary representations. This is suitable for our problem because we only focus on the prediction interval to reduce problem scales but do not need to predict an accurate value for continuous and integer variables. In this process, the integer values can potentially be negative. While two's complement is typically used to represent negative numbers in binary form (Baugh & Wooley, 1973), it is less intuitive for tasks that involve magnitude interpretation, such as logarithmic transformations. Instead, we introduce a sign bit $s \in \{0, 1\}$ to separately capture the sign of the variable, making the magnitude and sign easier to handle. Specifically, we record the optimal value $v_i$ of the variable $x_i$ in the historical instance, we calculate its logarithmic scale and binary sign bit $s$ as follows:

$$\mathbf{b}(v_i) = \text{bin}\left(\lfloor \log_2(|v_i| + 1)\rfloor\right), \quad s = \begin{cases} 0 & \text{if } v_i \le 10^{-5}, \\ 1 & \text{otherwise.} \end{cases}$$

where the vector $\mathbf{b}(v_i)$ represents the binary representation of the logarithmic value of $v_i$, prefixed by the sign bit $s$, and $\mathrm{bin}(\cdot)$ denotes the binary conversion.

# C. More details of thompson sampling algorithm

---

**Algorithm 1** Overall thompson sampling framework.

---

  **Input:**
  Predicted marginal probabilities $p_i$ for binary variables $x_i$,
  Predicted bounds for continuous/integer variables $x_j$.
  **Initialize prior distributions**:
  $\alpha_i, \beta_i \sim \mathrm{Beta}(p_i + 10^{-5}, 1 - p_i + 10^{-5})$ for binary variables,
  $\alpha_j, \beta_j \sim \mathrm{Beta}(1, 1)$ for continuous/integer variables.
  **for** each time step $t = 1, 2, \ldots$ **do**
    Sample $\mu_i \sim \mathrm{Beta}(\alpha_i, \beta_i)$ for all binary variables $x_i$
    Sample $\mu_j \sim \mathrm{Beta}(\alpha_j, \beta_j)$ for all continuous/integer variables $x_j$
    **Select Variables:**
    Rank and select the top $a\%$ for binary and continuous/integer variables
    **Fix Selected Variables:**
    For binary variables, fix values using Bernoulli distribution with probability $\mu_i$
    For continuous/integer variables, apply predicted bounds
    Solve subproblem with selected variable values to obtain solution $x_t$ and objective $z_t$
    **Update Parameters:**
    **if** $z_t$ is better than the best objective value $z^*$ from previous iterations **then**
      Update values of $\alpha$, $\beta$ for variables (detailed in Algorithm 2
      $z^* \leftarrow z_t$
    **end if**
    **if** solution becomes infeasible **then**
      Apply relaxation mechanism (detailed in Algorithm 3
    **end if**
  **end for**

---

The Parameter update process algorithm is designed to update the parameters of the pior distributions for binary and continuous/integer variables based on the outcomes of the current solution. The goal is to refine these parameters to improve the performance of the Thompson Sampling approach in subsequent iterations. We adjust our fixing strategy using the Beta distribution parameters, $\alpha$ and $\beta$. The mean of the $Beta(\alpha, \beta)$ distribution is $\frac{\alpha}{\alpha+\beta}$. As these parameters increase, the distribution becomes more concentrated around the mean. With a prior of $Beta(\alpha, \beta)$, the posterior updates to $Beta(\alpha + 1, \beta)$ or $Beta(\alpha, \beta + 1)$.

In each iteration, we fix a percentage a% of the variables. When we find a better solution, we update the Beta distribution for the remaining 1-a% of unfixed variables based on this new solution. We also compare the current strategy to the one from the previous round that gave the best solution. If a variable was fixed before but left unfixed in the current iteration, it indicates the previous strategy limited the solution quality. We update the Beta distributions for these variables to reflect this. In the next round, we resample the fixing strategy using these updated Beta distributions.

**Update $\alpha$, $\beta$ for binary variables.** For binary variables $x_i, i \in \{1, 2, \ldots, p\}$, we rank variables based on $\min(\mu_i, 1 - \mu_i)$, selecting the lowest $a\%$, and sample fixed values from the Bernoulli distribution with probability $\mu_i$. At each iteration $t$, the priors for unselected binary variables are updated based on their observed outcomes: For unselected binary variables, we set $\alpha_i = \alpha_i + 1$ when $x_i = 1$, and set $\beta_i = \beta_i + 1$ when $x_i = 0$. For selected binary variables, if the current solution $x_t$ is better than the previous best $x^*_{t-1}$, compare the set of selected variables $a_t$ with the previously best set $a^*_{t-1}$. For variables where the current value is 1 but was 0 in $a^*_{t-1}$ (or was not selected), we set $\alpha_i = \alpha_i + 1$. If the current value is 0, and it was 1 in $a^*_{t-1}$ (or was not selected), we set $\beta_i = \beta_i + 1$.

**Update $\alpha$, $\beta$ for integer and continuous variables.** For integer and continuous variables, we select the top $a\%$ of these variables based on their $\mu_j$ values and apply the predicted upper and lower bounds. The priors for continuous/integer variables are updated as follows: For unselected variables, if the variable's actual value in the solution falls within the

predicted bounds, we set $\alpha_j = \alpha_j + 1$. If it does not satisfy the predicted bounds, we set $\beta_j = \beta_j + 1$. For selected variables, when the current solution $x_t$ is better than the previous best $x_{t-1}^*$ (i.e., $r_t = 1$), we compare the set of selected variables $a_t$ and the previously best set $a_{t-1}^*$. If $x_j$ was unselected in $a_t$ but was selected in $a_{t-1}^*$, we set $\beta_j = \beta_j + 1$. In contrast, if a variable $x_j$ was selected in $a_t$ but not in $a_{t-1}^*$, no immediate conclusion about its benefit can be drawn, since the objective is to rule out incorrect bound predictions.

---

**Algorithm 2** The parameter update process algorithm.

---

**Input:** Current solution $x_t$, Best solution $x^*$ from previous iterations, Best objective value $z^*$

**if** $z_t$ is better than $z^*$ **then**

    Set $r_t = 1$

    Update the best solution $x_t^* = x_t$

    Update priors for binary variables $i$:

    **for** each unselected binary variable $i$ **do**

        **if** $x_i = 1$ **then**

            $\alpha_i \leftarrow \alpha_i + 1$

        **else**

            $\beta_i \leftarrow \beta_i + 1$

        **end if**

    **end for**

    **for** each selected binary variable $i$ **do**

        **if** $x_i = 1$ and $x^* = 0$ **then**

            $\alpha_i \leftarrow \alpha_i + 1$

        **else if** $x_i = 0$ and $x^* = 1$ **then**

            $\beta_i \leftarrow \beta_i + 1$

        **end if**

    **end for**

    Update priors for continuous/integer variables $j$:

    **for** each unselected continuous/integer variable $j$ **do**

        **if** $x_j$ violates predicted bounds **then**

            $\beta_j \leftarrow \beta_j + 1$

        **else**

            $\alpha_j \leftarrow \alpha_j + 1$

        **end if**

    **end for**

    **for** each selected continuous/integer variable $j$ **do**

        **if** $x_j$ wasn't selected in the previous best solution $x_{t-1}^*$ **then**

            $\beta_j \leftarrow \beta_j + 1$

        **end if**

    **end for**

  **else**

    Set $r_t = 0$

**end if**

---

## D. More details of relaxation mechanism

When faced with infeasible instances, it typically indicates that some variable predictions are incorrect, resulting in conflicts with constraints. Our relaxation mechanism addresses this by dividing the conflicting variables into $G$ groups and subsequently solving each without these variable sets. We choose $G = 10$ in our evaluation.

To further determine the variables' feasible ranges, the upper and lower bounds are converted back into their corresponding integer forms, denoted as $k_{ub}$ and $k_{lb}$. For positive variables ($s = 1$), we represent the variable's value of the optimal solution in the form $2^k + m$, where $k \geq 0, 0 \leq m \leq 2^k - 1$. From the inequality $k < \lfloor \log_2(|v_i| + 1) \rfloor \leq k + 1$, the predicted range for the variable lies between $2^{k_{lb}} - 1$ and $2^{k_{ub}+1}$. For negative variables ($s = 0$), the ranges are symmetrically calculated, spanning from $-2^{k_{ub}+1}$ to $-2^{k_{lb}} + 1$. For continuous variables, we first round them to the nearest integer and then process

them similarly to integer variables.

When a feasible solution cannot be found, we repeatedly apply the relaxation mechanism, building upon previous relaxations. Each iteration of this mechanism reduces the number of fixed variables. Therefore, theoretically, with enough iterations, we can ensure that the variables causing conflicts with the constraints are filtered out. However, in practice, it usually takes only a few iterations to obtain a feasible solution. For example, in the case of bnd_1, there are errors for only 8 for 2993 binary variables. By splitting these 8 erroneous variables into 10 groups, at least one group will inevitably exclude the erroneous variables. Of course, when the number of erroneous variables is greater, this is not guaranteed, but it is important to note that some variables, even if mispredicted, do not affect the ability to find a feasible solution due to their limited impact on solution sensitivity. We can easily filter out some variables that are highly sensitive to solution quality for each group.

---

**Algorithm 3** Relaxation Mechanism

---

   **if** solution becomes infeasible **then**
      Divide fixed variables into $G$ groups.
      **for** each group $g = 1, 2, \ldots, G$ **do**
         Relax fixed variables in group $g$ back to their original bounds.
         Solve the subproblem with these relaxed constraints.
         **if** feasible solution found **then**
            Proceed to the next iteration.
         **end if**
      **end for**
   **end if**

---

# E. More details of experiments

## E.1. Datasets

We selected the datasets based on two key considerations: first, the varying components within the instances, and second, the number of different variable types (integer, binary, continuous) present in each dataset. We aim for our evaluation to cover a wide range of variable types and varying components as comprehensively as possible.

The varying components of nine datasets are summarized in Table 6.

*Table 6.* The varying components of datasets.

| Datasets | Varying component | | | | | |
|---|---|---|---|---|---|---|
| | LO | UP | OBJ | LHS | RHS | MAT |
| bnd_1 | ✓ | | | | | |
| bnd_2 | ✓ | ✓ | | | | |
| bnd_3 | ✓ | ✓ | | | | |
| mat_1 | | | | | | ✓ |
| obj_1 | | | ✓ | | | |
| obj_2 | | | ✓ | | | |
| rhs_1 | | | | ✓ | ✓ | |
| rhs_2 | | | | | ✓ | |
| rhs_3 | | | | | ✓ | |

The details of each dataset is as follows:

**bnd_1:** This dataset is from "bnd_s1" in the MIP Computational Competition 2023 (Bolusani et al., 2023). The instance is based on the instance *rococoC10-001000* from the MIPLIB 2017 benchmark library (Gleixner et al., 2021). The instances were generated by perturbing the upper bounds of general integer variables selected via a discrete uniform distribution up to ±100% of the bound value.

**bnd_2:** This dataset is from "bnd_s2" in the MIP Computational Competition 2023 (Bolusani et al., 2023). This series is based on the instance *csched007* from the MIPLIB 2017 benchmark library (Gleixner et al., 2021). The instances were generated via random fixings of 15% to 25% of the binary variables selected via a discrete uniform distribution w.r.t. the original instance.

**bnd_3:** This dataset is from "bnd_s3" in the MIP Computational Competition 2023 (Bolusani et al., 2023). This series is also based on the instance *csched007* from the MIPLIB 2017 benchmark library (Gleixner et al., 2021). The instances were generated via random fixings of 5% to 20% of the binary variables selected via a discrete uniform distribution w.r.t. the original instance. These instances are relatively harder to solve as compared to the instances in **bnd_2**.

**mat_1:** This dataset is from "mat_s1" in the MIP Computational Competition 2023 (Bolusani et al., 2023). This series is based on the optimal vaccine allocation problem (Tanner & Ntaimo, 2010) and generated with varying constraint coefficients in the inequality constraints.

**obj_1:** This dataset is from "obj_s1" in the MIP Computational Competition 2023 (Bolusani et al., 2023). This series is based on the stochastic multiple binary knapsack problem (Angulo et al., 2016). The problem is modeled as a two-stage stochastic MILP and one-third of the objective vector varying across instances.

**obj_2:** This dataset is from "obj_s2" in the MIP Computational Competition 2023 (Bolusani et al., 2023). The instances are based on the instance *ci-s4* from the MIPLIB 2017 benchmark library (Gleixner et al., 2021) with random perturbations and random rotations of the objective vector.

**rhs_1:** This dataset is from "rhs_s1" in the MIP Computational Competition 2023 (Bolusani et al., 2023). This series is based on the stochastic server location problem (Ntaimo, 2010). The instances is generated by the given dataset, and only the right-hand side vector of equality constraints varying across instances.

**rhs_2:** This dataset is from "rhs_s2" in the MIP Computational Competition 2023 (Bolusani et al., 2023). This series is based on a synthetic MILP and the associated dataset (Jiménez-Cordero et al., 2022). The instances are generated by taking a convex combination of two different RHS vectors.

**rhs_3:** This dataset is from "rhs_s4" in the MIP Computational Competition 2023 (Bolusani et al., 2023). This series is also based on the synthetic MILP (Jiménez-Cordero et al., 2022). The instances are generated by taking a convex combination of two different RHS vectors(different than the ones used for generating **rhs_2**).

### E.2. Implementation details of the baselines

All baselines that provided open-source implementations, including PS and Re_Tuning, were tested using their official code. Since ND did not provide open-source code, we reproduced their method to the best of our ability based on their paper (Nair et al., 2020) and fine-tuned the parameters accordingly.

**SCIP.** We use SCIP 8.0.4 (Bestuzheva et al., 2021), which is the state-of-the art open source solver. We keep all the other SCIP parameters to default and emphasize that all of the SCIP solver's advanced features, such as presolve and heuristics, are open.

**Re_Tuning.** Re_Tuning is a state-of-the-art heuristic reoptimization framework (Patel, 2024), which does not utilize machine learning models. This framework, developed for the MIP 2023 workshop's computational competition (Bolusani et al., 2023), earned the first prize. It is primarily based on reusing historical branches and fine-tuning SCIP's parameters for more effective reoptimization. Our investigation revealed that Re_Tuning adjusts its configurations based on the previous instances it solves. Specifically, it may disable modules such as presolving or generating cutting planes for subsequent instances. While these adjustments have been shown to potentially improve overall solving time on certain datasets, they inevitably make it more challenging to find high-quality feasible solutions quickly in the early stages. To address this, we ensured these modules remained enabled for all instances, striving to achieve the best possible results with their code.

**Predict-and-Search(PS).** PS is an end-to-end machine learning-based approach (Han et al., 2023) which employs large neighborhood search (LNS) combined with GNN predictions. In practice, we do not know how many variables may be predicted incorrectly, and selecting an appropriate radius $\delta$ for the neighborhood in LNS can be time-consuming. To better demonstrate the performance of the PS method, we select the radius $\delta$ based on the average number of binary prediction errors observed during our preliminary tests, as shown in Table 1.

**Neural Diving(ND).** Another notable method we compared against is Neural Diving (**ND**) framework with Selective

Net (Nair et al., 2020), which is also based on a variable-fixing strategy. Since ND focuses on fixing variables to accelerate the solving process, it serves as a relevant baseline to evaluate alongside our approach.

### E.3. More results with different parameters

In this section, we present a comprehensive evaluation of policy performance across various synthetic and real-world datasets, using different time and fix parameters. Each table below illustrates the impact of varying these parameters on the performance metrics, namely the absolute and relative gaps. The methods examined include SCIP, Re_Tuning, ND, and PS, alongside our proposed method, VP-OR, under different time constraints and fixed parameter ratios.

Table 7 illustrates the performance of various methods under different boundary conditions (bnd_1, bnd_2, bnd_3). After reoptimizing with adjusted boundary parameters, the VP-OR method consistently shows lower absolute and relative gaps compared to SCIP and other comparative methods under different time constraints (T=10 and T=20).

*Table 7.* Policy evaluation on the synthetic and real-world datasets with different time and fix parameters. We report the arithmetic mean of gap_abs and gap_rel.

| Methods | bnd_1 | | bnd_2 | | bnd_3 | |
|---|---|---|---|---|---|---|
| | gap_abs | gap_rel | gap_abs | gap_rel | gap_abs | gap_rel |
| SCIP (T=10.0) | 1974.20 | 0.16 | - | - | - | - |
| Re_Tuning (T=10.0) | 1425.5 | 0.12 | - | - | - | - |
| ND (T=10.0, P=0.5) | - | - | - | - | - | - |
| PS (T=10.0, P=0.5) | 9439.60 | 0.79 | - | - | - | - |
| VP-OR(Ours) (T=10.0, P=0.5) | 279.20 | 0.02 | 40.20 | 0.11 | 31.20 | 0.09 |
| ND (T=10.0, P=0.6) | - | - | - | - | - | - |
| PS (T=10.0, P=0.6) | 9439.60 | 0.79 | - | - | - | - |
| VP-OR(Ours) (T=10.0, P=0.6) | 528.80 | 0.04 | 39.40 | 0.11 | 37.40 | 0.11 |
| ND (T=10.0, P=0.7) | - | - | - | - | - | - |
| PS (T=10.0, P=0.7) | 9665.20 | 0.81 | - | - | - | - |
| VP-OR(Ours) (T=10.0, P=0.7) | 299.40 | 0.02 | 40.20 | 0.11 | 28.60 | 0.06 |
| ND (T=10.0, P=0.8) | - | - | - | - | - | - |
| PS (T=10.0, P=0.8) | 1216.80 | 0.10 | - | - | - | - |
| VP-OR(Ours) (T=10.0, P=0.8) | 973.80 | 0.08 | 38.80 | 0.11 | 34.80 | 0.10 |
| SCIP (T=20.0) | 921.00 | 0.08 | - | - | - | - |
| Re_Tuning (T=20.0) | 402.25 | 0.03 | 26.0 | 0.06 | - | - |
| ND (T=20.0, P=0.5) | - | - | - | - | - | - |
| PS (T=20.0, P=0.5) | 2483.00 | 0.20 | - | - | - | - |
| VP-OR(Ours) (T=20.0, P=0.5) | 313.20 | 0.03 | 48.60 | 0.14 | 33.80 | 0.10 |
| ND (T=20.0, P=0.6) | - | - | - | - | - | - |
| PS (T=20.0, P=0.6) | 2408.00 | 0.19 | - | - | - | - |
| VP-OR(Ours) (T=20.0, P=0.6) | 264.60 | 0.02 | 40.40 | 0.12 | 37.40 | 0.11 |
| ND (T=20.0, P=0.7) | - | - | - | - | - | - |
| PS (T=20.0, P=0.7) | 2627.40 | 0.21 | - | - | - | - |
| VP-OR(Ours) (T=20.0, P=0.7) | 299.40 | 0.02 | 39.80 | 0.11 | 23.40 | 0.07 |
| ND (T=20.0, P=0.8) | - | - | - | - | - | - |
| PS (T=20.0, P=0.8) | 1007.20 | 0.08 | - | - | - | - |
| VP-OR(Ours) (T=20.0, P=0.8) | 1409.40 | 0.12 | 41.40 | 0.12 | 23.40 | 0.07 |

Table 8 evaluates performance under different matrix and objective function settings (mat_1, obj_1, obj_2). With these adjustments, the VP-OR method maintains significant suppression of gap_abs and gap_rel, particularly excelling in objective function cases (obj_1 and obj_2).

*Table 8.* Policy evaluation on the synthetic and real-world datasets with different time and fix parameters. We report the arithmetic mean of gap_abs and gap_rel.

| Methods | mat_1 | | obj_1 | | obj_2 | |
|---|---|---|---|---|---|---|
| | gap_abs | gap_rel | gap_abs | gap_rel | gap_abs | gap_rel |
| SCIP (T=10.0) | 14.10 | 0.23 | 11.40 | 0.00 | 626.52 | 0.39 |
| Re_Tuning (T=10.0) | 30.06 | 0.48 | 10.25 | 0.00 | 74.10 | 0.09 |
| ND (T=10.0, P=0.5) | - | - | 11.40 | 0.00 | 634.70 | 0.39 |
| PS (T=10.0, P=0.5) | 14.10 | 0.23 | 13.40 | 0.00 | 387.89 | 0.51 |
| VP-OR(Ours) (T=10.0, P=0.5) | 9.09 | 0.15 | - | - | 7783.94 | 1.53 |
| ND (T=10.0, P=0.6) | - | - | 11.40 | 0.00 | 634.70 | 0.39 |
| PS (T=10.0, P=0.6) | 14.10 | 0.23 | 13.40 | 0.00 | 397.53 | 0.51 |
| VP-OR(Ours) (T=10.0, P=0.6) | 11.62 | 0.19 | - | - | 6854.66 | 0.75 |
| ND (T=10.0, P=0.7) | - | - | 11.40 | 0.00 | 674.21 | 0.44 |
| PS (T=10.0, P=0.7) | 14.10 | 0.23 | 13.40 | 0.00 | 397.53 | 0.51 |
| VP-OR(Ours) (T=10.0, P=0.7) | 10.09 | 0.16 | 3.28 | 0.00 | 329.99 | 0.06 |
| ND (T=10.0, P=0.8) | - | - | 11.40 | 0.00 | - | - |
| PS (T=10.0, P=0.8) | 14.10 | 0.23 | 13.40 | 0.00 | 702.68 | 0.41 |
| VP-OR(Ours) (T=10.0, P=0.8) | 11.77 | 0.19 | 338.60 | 0.04 | 8287.57 | 3.01 |
| SCIP (T=20.0) | 11.66 | 0.19 | 10.40 | 0.00 | 285.99 | 0.14 |
| Re_Tuning (T=20.0) | 18637.00 | 0.42 | 8.25 | 0.00 | 1.61 | 0.01 |
| ND (T=20.0, P=0.5) | - | - | 10.40 | 0.00 | 285.99 | 0.14 |
| PS (T=20.0, P=0.5) | 13.17 | 0.21 | 13.40 | 0.00 | 243.40 | 0.30 |
| VP-OR(Ours) (T=20.0, P=0.5) | 7.69 | 0.12 | - | - | 6855.85 | 0.75 |
| ND (T=20.0, P=0.6) | - | - | 10.40 | 0.00 | 268.18 | 0.13 |
| PS (T=20.0, P=0.6) | 13.17 | 0.21 | 13.40 | 0.00 | 243.40 | 0.30 |
| VP-OR(Ours) (T=20.0, P=0.6) | 9.94 | 0.16 | 19.40 | 0.00 | 6058.61 | 0.50 |
| ND (T=20.0, P=0.7) | - | - | 10.40 | 0.00 | 285.99 | 0.14 |
| PS (T=20.0, P=0.7) | 13.17 | 0.21 | 13.40 | 0.00 | 239.79 | 0.28 |
| VP-OR(Ours) (T=20.0, P=0.7) | 10.09 | 0.16 | 3.28 | 0.00 | 12.85 | 0.01 |
| ND (T=20.0, P=0.8) | - | - | 10.40 | 0.00 | - | - |
| PS (T=20.0, P=0.8) | 13.17 | 0.21 | 13.40 | 0.00 | 202.44 | 0.16 |
| VP-OR(Ours) (T=20.0, P=0.8) | 11.61 | 0.19 | 142.40 | 0.02 | 12.85 | 0.01 |

Table 9 shows the response of each method when adjusting the parameters on the right-hand side of constraints (rhs_1, rhs_2, rhs_3). In these scenarios, the VP-OR method achieves gaps close to zero.

*Table 9.* Policy evaluation on the synthetic and real-world datasets with different time and fix parameters. We report the arithmetic mean of gap_abs and gap_rel.

| Methods | rhs_1 | | rhs_2 | | rhs_3 | |
|---|---|---|---|---|---|---|
| | gap_abs | gap_rel | gap_abs | gap_rel | gap_abs | gap_rel |
| SCIP (T=10.0) | 173.08 | 0.50 | 12.29 | 0.00 | 16.77 | 0.00 |
| Re_Tuning (T=10.0) | 6.40 | 0.02 | 2.24 | 0.00 | 0.40 | 0.00 |
| ND (T=10.0, P=0.5) | - | - | - | - | - | - |
| PS (T=10.0, P=0.5) | 57558.07 | 165.41 | 13.23 | 0.00 | 12.46 | 0.00 |
| VP-OR(Ours) (T=10.0, P=0.5) | 0.27 | 0.00 | 1.85 | 0.00 | 0.26 | 0.00 |
| ND (T=10.0, P=0.6) | - | - | - | - | - | - |
| PS (T=10.0, P=0.6) | 62046.33 | 177.93 | 13.23 | 0.00 | 12.46 | 0.00 |
| VP-OR(Ours) (T=10.0, P=0.6) | 0.50 | 0.00 | 1.85 | 0.00 | 0.26 | 0.00 |
| ND (T=10.0, P=0.7) | - | - | - | - | - | - |
| PS (T=10.0, P=0.7) | 67090.50 | 193.04 | 22.25 | 0.00 | 17.98 | 0.00 |
| VP-OR(Ours) (T=10.0, P=0.7) | 0.73 | 0.00 | 1.85 | 0.00 | 0.26 | 0.00 |
| ND (T=10.0, P=0.8) | - | - | - | - | - | - |
| PS (T=10.0, P=0.8) | 66978.45 | 192.33 | 15.35 | 0.00 | 16.29 | 0.00 |
| VP-OR(Ours) (T=10.0, P=0.8) | 0.71 | 0.00 | 1.85 | 0.00 | 0.26 | 0.00 |
| SCIP (T=20.0) | 173.08 | 0.50 | 5.54 | 0.00 | 7.22 | 0.00 |
| Re_Tuning (T=20.0) | 2.85 | 0.01 | 0.00 | 0.00 | 0.00 | 0.00 |
| ND (T=20.0, P=0.5) | - | - | - | - | - | - |
| PS (T=20.0, P=0.5) | 38275.26 | 109.85 | 5.54 | 0.00 | 5.97 | 0.00 |
| VP-OR(Ours) (T=20.0, P=0.5) | 0.39 | 0.00 | 1.85 | 0.00 | 0.26 | 0.00 |
| ND (T=20.0, P=0.6) | - | - | - | - | - | - |
| PS (T=20.0, P=0.6) | 38275.26 | 109.85 | 5.47 | 0.00 | 5.97 | 0.00 |
| VP-OR(Ours) (T=20.0, P=0.6) | 0.26 | 0.00 | 1.85 | 0.00 | 0.26 | 0.00 |
| ND (T=20.0, P=0.7) | - | - | - | - | - | - |
| PS (T=20.0, P=0.7) | 65141.18 | 187.43 | 4.42 | 0.00 | 5.97 | 0.00 |
| VP-OR(Ours) (T=20.0, P=0.7) | 0.29 | 0.00 | 1.85 | 0.00 | 0.26 | 0.00 |
| ND (T=20.0, P=0.8) | - | - | - | - | - | - |
| PS (T=20.0, P=0.8) | 54004.10 | 155.03 | 3.30 | 0.00 | 2.16 | 0.00 |
| VP-OR(Ours) (T=20.0, P=0.8) | 0.40 | 0.00 | 1.85 | 0.00 | 0.26 | 0.00 |

### E.4. More results of the relative gap

In the main text, we presented results for the datasets bnd_1, mat_1, and rhs_1. Here, we extend our analysis by providing additional results for the remaining datasets. This section focuses on performance comparisons in terms of the average relative gap *gap_rel*.

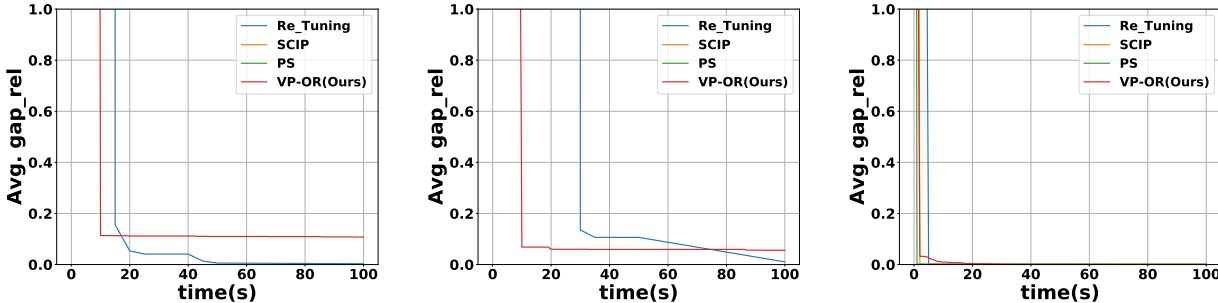

*Figure 3.* Performance comparisons in bnd_2, bnd_3 and obj_1, where the y-axis is average relative primal gap; each plot represents one benchmark dataset.

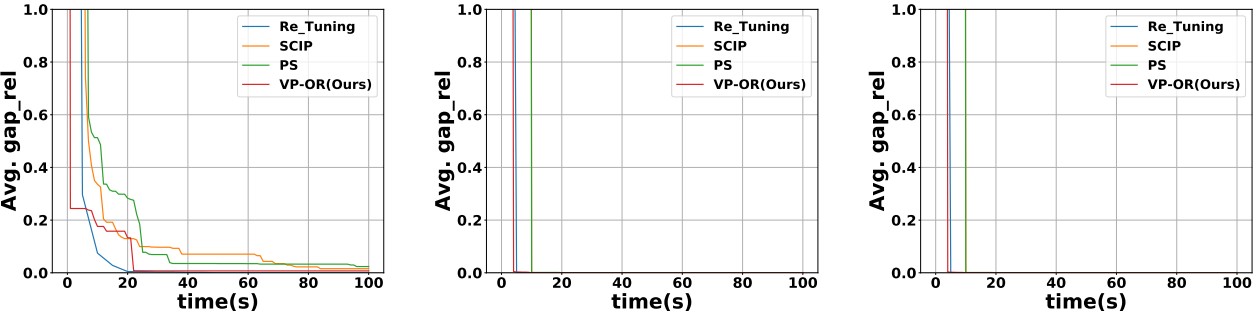

*Figure 4.* Performance comparisons in obj_2, rhs_2 and rhs_3, where the y-axis is average relative primal gap; each plot represents one benchmark dataset.

### E.5. Ablation study of prediction.

The table below demonstrates the predictive performance of both traditional Graph Neural Networks (GNN) and our approach in a reoptimization context(Re_GNN):

### E.6. Computational complexity analysis

The primary computational complexity of VP-OR arises from the Thompson Sampling process. In the sampling phase of Thompson Sampling, we sample the probability $p$ for binary variables and select a certain percentage (a%) of variables based on the value of $min(p, 1 − p)$ by sorting them. For integer and continuous variables, we sample to determine whether they should be fixed and select the top a% of variables based on this criterion. This step has a time complexity of $O(nlogn)$, where n is the number of variables.

In the parameter update phase of Thompson Sampling, we update the parameters for each variable once. This step has a time complexity of $O(n)$. We tested the sampling time and parameter update time for each dataset, as presented in Table 11.

*Table 10.* Predictive performance of traditional Graph Neural Networks (GNN) and our prediction method in a reoptimization context(Re_GNN).

|  | bnd_1 | bnd_2 | bnd_3 |
|---|---|---|---|
| Total binary var. | 1457.0 | 1457.0 | 1457.0 |
| mispredicted binary var. (GNN) | 163.0 | 45.1 | 42.2 |
| mispredicted binary var. (Re_GNN) | **8.2** | **6.7** | **4.5** |
| Total integer var. | 124.0 | 0.0 | 0.0 |
| mispredicted integer var. (GNN) | 33.4 | 0.0 | 0.0 |
| mispredicted integer var. (Re_GNN) | **17.4** | 0.0 | 0.0 |
| Total continuous var. | 0.0 | 301.0 | 301.0 |
| mispredicted continuous var. (GNN) | 0.0 | 140.2 | 121.0 |
| mispredicted continuous var. (Re_GNN) | **0.0** | **0.0** | **2.0** |

*Table 11.* Variable Numbers, Sampling Time(Time_s), and Parameter Update Time(Time_u) for Different Datasets

|  | bnd_1 | bnd_2 | bnd_3 | mat_1 | obj_1 | obj_2 | rhs_1 | rhs_2 | rhs_3 |
|---|---|---|---|---|---|---|---|---|---|
| **Var. num** | 3117 | 1758 | 1758 | 802 | 360 | 745 | 12760 | 1000 | 1000 |
| **Time_s (s)** | 0.008 | 0.005 | 0.003 | 0.013 | 0.002 | 0.016 | 0.102 | 0.002 | 0.002 |
| **Time_u (s)** | 0.002 | 0.001 | 0.000 | 0.009 | 0.001 | 0.010 | 0.070 | 0.001 | 0.001 |

## E.7. Results for the impact of initial hints

We provided the initial solution as a hint for the "completesol" heuristic method during the presolve phase, effectively employing the historical solution as a warm start(WS). We observe that SCIP has improvements in the quality of feasible solutions under these conditions. The results are shown in Table 12, where "-" indicates cases where the method could not find a feasible solution within the designated time limit.

*Table 12.* Performance Comparison Across SCIP, SCIP(WS) and VP-OR. We report the arithmetic mean of the metric gap_rel.

| Method | bnd_1 | bnd_2 | bnd_3 | mat_1 | obj_1 | obj_2 | rhs_1 | rhs_2 | rhs_3 |
|---|---|---|---|---|---|---|---|---|---|
| SCIP | 0.16 | - | - | 0.23 | 0.00 | 0.39 | 0.50 | 0.00 | 0.00 |
| SCIP(WS) | 0.10 | - | - | 0.22 | 0.00 | 0.12 | 0.50 | 0.00 | 0.00 |
| VP-OR(Ours) | **0.02** | **0.11** | **0.06** | **0.16** | **0.00** | **0.06** | **0.00** | **0.00** | **0.00** |

## E.8. More results for expanded test samples

The publicly available dataset from the MIP Workshop 2023 Computational Competition on Reoptimization (Bolusani et al., 2023) is limited in size, providing only 50 examples per task. To further increase the number of test samples, we attempt to generate similar datasets for testing by using a method consistent with the one published by the competition organizers. This step proves to be very time-consuming because random perturbations in the parameters often result in infeasible problems. During the dataset generation process, we repeatedly generate instances randomly until we find one that is feasible. Using the bnd_1 dataset as an example, we generate 100 additional instances. The results presented in Table 13 are consistent with our previous tests.

## E.9. More results for end-to-end methods

Several end-to-end methods have been developed specifically for large-scale problems, such as GNN&GBDT (Ye et al., 2023) and Light-MILPopt (Ye et al., 2024). We conduct an experiment on the latest approach, Light-MILPopt. We observe that Light-MILPopt uses a variable fixing strategy, initially fixing k% of the variables based on predicted values (using the default setting k=20 as per the authors' code). However, in a reoptimization context, fixing these variables often led to

*Table 13.* Policy evaluation on the bnd_1 dataset with 100 samples. We provide the metrics Average Relative Gap (gap_rel) and Average Absolute Gap (gap_abs).

| Method | gap_rel | gap_abs |
|---|---|---|
| SCIP | 0.20 | 2354.2 |
| PS | 1.13 | 13213.0 |
| VP-OR(Ours) | **0.01** | **167.3** |

infeasibility in most instances. This is mainly because the model inaccurately predicts some variables, even when considered high-confidence. Consequently, we test the results with the variable fixing module disabled. The final experimental results present the number of instances that can find feasible solutions within a 10-second time limit in Table 14. Although this method is not specifically designed for reoptimization scenarios, which often demand rapid responses to slight changes in parameters with time-critical requirements for solutions, it does show some improvement over SCIP on more challenging datasets like bnd_2 and bnd_3.

*Table 14.* Number of Instances Finding Feasible Solutions within 10 Seconds.

| Method | bnd_1 | bnd_2 | bnd_3 |
|---|---|---|---|
| SCIP | 5/5 | 0/5 | 0/5 |
| Light-MILPopt | 0/5 | 0/5 | 0/5 |
| Light-MILPopt (without fix strategy) | 5/5 | 1/5 | 1/5 |
| VP-OR (Ours) | **5/5** | **5/5** | **5/5** |

We provide the average relative gap (gap_rel) for comparison in Table 15, where "-" represents cases where the method could not find a feasible solution within the time limit.

*Table 15.* Average Relative Gap (gap_rel).

| | bnd_1 | bnd_2 | bnd_3 |
|---|---|---|---|
| SCIP | 0.16 | - | - |
| Light-MILPopt | - | - | - |
| Light-MILPopt (without fix strategy) | 0.22 | - | - |
| VP-OR (Ours) | **0.02** | **0.11** | **0.06** |

### E.10. Large-scale MILP experiments

We expand our experiments to include large-scale MILP experiments with more instances (200 for training and 100 for testing) and add a comparison with the ML-guided LNS method of Huang et al. (ConPas) (Huang et al., 2024). The results are reported for both Gurobi and SCIP. We generate large-scale datasets IS and CA using the code from Gasse et al. (Gasse et al., 2019), consistent with those used by Han et al. The Gurobi and SCIP solvers could not reach an optimal solution within 3600 seconds for these instances. We run Gurobi for 3600 seconds to record the incumbent solution. For evaluation, we impose a 30-second time limit and update the incumbent solution if a better one is found. Fig. 16 and Fig. 17 shows that VP-OR achieves more significant acceleration than PS (Han et al., 2023) and ConPas (Huang et al., 2024) during the early stages of solving on both Gurobi and SCIP.

### E.11. Ablation experiments

We conduct an ablation experiment in Fig. 18 using Re-GNN for initial solution prediction in LNS and traditional GNN with the Thompson Sampling (TS) algorithm. We report the average gap_rel, where "-" means no feasible solution was found.

*Table 16.* Average absolute gap (gap_abs), relative gap (gap_rel) and the number of each method that achieves the closest solution to the optimal one within the 30-second time limit (Wins) of the CA dataset.

|  | gap_abs | gap_rel | Wins |
|---|---|---|---|
| SCIP | 24068.27 | 0.19 | 0/100 |
| Gurobi | 3754.16 | 0.03 | 0/100 |
| PS+SCIP | 20182.91 | 0.19 | 0/100 |
| PS+Gurobi | 3754.16 | 0.03 | 0/100 |
| ConPaS+SCIP | 8506.52 | 0.07 | 0/100 |
| ConPaS+Gurobi | 2526.28 | 0.02 | 0/100 |
| VP-OR+SCIP(Ours) | 0.00 | 0.00 | 12/100 |
| VP-OR+Gurobi(Ours) | **0.00** | **0.00** | **88/100** |

*Table 17.* Average absolute gap (gap_abs), relative gap (gap_rel) and the number of each method that achieves the closest solution to the optimal one within the 30-second time limit (Wins) of the IS dataset.

|  | gap_abs | gap_rel | Wins |
|---|---|---|---|
| SCIP | 104.92 | 0.05 | 0/100 |
| Gurobi | 133.07 | 0.06 | 0/100 |
| PS+SCIP | 104.57 | 0.05 | 0/100 |
| PS+Gurobi | 117.62 | 0.06 | 0/100 |
| ConPaS+SCIP | 30.25 | 0.01 | 0/100 |
| ConPaS+Gurobi | 24.66 | 0.01 | 0/100 |
| VP-OR+SCIP(Ours) | 9.57 | 0.00 | 33/100 |
| VP-OR+Gurobi(Ours) | **0.97** | **0.00** | **67/100** |

Re-GNN improves performance over traditional GNN in the LNS framework. The method combining traditional GNN with TS also quickly finds feasible solutions. The combined approach, VP-OR, achieves the best results across all datasets.

*Table 18.* Average absolute gap (gap_abs) within the 10-second time limit of the datasets, where "-" means no feasible solution was found.

|  | bnd_1 | bnd_2 | bnd_3 | mat_1 | obj_1 | obj_2 | rhs_1 | rhs_2 | rhs_3 |
|---|---|---|---|---|---|---|---|---|---|
| GNN+LNS(PS) | 0.81 | - | - | 0.23 | 0.00 | 0.51 | 193.04 | 0.00 | 0.00 |
| Re-GNN+LNS | 0.16 | - | - | 0.23 | 0.00 | 0.21 | 1.04 | 0.00 | 0.00 |
| GNN+TS | 0.04 | 0.17 | 0.12 | 0.17 | 0.00 | 0.00 | 0.00 | 0.00 | 0.00 |
| Re-GNN+TS(VP-OR) | 0.02 | 0.11 | 0.06 | 0.16 | 0.00 | 0.00 | 0.00 | 0.00 | 0.00 |

### E.12. Accuracy of predicted bounds for integer and continuous variables

We evaluate the accuracy of predicted bounds for integer and continuous variables as follows: if the optimal value of a variable lies within the predicted lower and upper bounds, the predicted bounds are considered accurate; otherwise, they are not. Continuous and integer variables are treated similarly, with the key difference being that continuous variables are rounded during preprocessing.

In Fig. 19, we evaluate how tight these bounds become after training and across different optimization stages, we use I_o and C_o to denote the average difference between original upper and lower bounds for integer and continuous variables, respectively. I_p and C_p represent the average predicted bounds gap for integers and continuous variables. In cases where there are no integer or continuous variables in a dataset, we use "NA" as a placeholder.

*Table 19.* Average difference between original and predicted bounds of the datasets, where "NA" means there are no integer or continuous variables in the dataset.

|  | bnd_1 | bnd_2 | bnd_3 | mat_1 | obj_2 | rhs_1 | rhs_2 | rhs_3 |
|---|---|---|---|---|---|---|---|---|
| I_o | 72146.14 | NA | NA | NA | 2.0 | NA | NA | NA |
| I_p | 32750.57 | NA | NA | 1.34 | NA | NA | NA | NA |
| C_o | NA | 25.91 | 26.15 | 1.0 | 1e+19 | 1e+20 | 2e+20 | 2e+20 |
| C_p | NA | 14.22 | 14.28 | 1.0 | 19968.14 | 1.0 | 14.22 | 13.67 |

