# OpenReview forum: "Don't Restart, Just Reuse: Reoptimizing MILPs with Dynamic Parameters"
_ICML.cc/2025/Conference — ICML 2025 poster_

### Official Review · Reviewer_L6hn · 2025-03-13

**Overall Recommendation:** 4

**Summary:**

The paper presents a new heuristic to find primal solutions of mixed-integer linear problems (MILP). The approach is based on predicting a distribution for the value of binary variables and a multi-armed bandit approach for iterative variable fixing. The paper also considers new features for the graph-embedding of MILPs based on values at leaf nodes of historical optimal solutions. Experiments on instances of the 2023 MIP Competition measure the time to find a feasible solution as well as the relative gap for the method and several baselines.

**Claims And Evidence:**

The paper claims to provide significant improvement in times to find feasible solutions. I have major concerns regarding the experiment setting and overall presentation of the results, which follows from an ambiguous framing of the paper’s goal and results.

First, two of the four baselines are general MILP solvers (SCIP and Re_Tuning). These methods are not only aiming to find feasible solutions, but *proving their global optimality*. It is well-known that the latter is a much harder task than the former. The comparison with these two baselines is biased since it is assumed that the optimal value OPT is known in advance, but SCIP and Re_Tuning are spending the majority of their computational budget to find OPT and prove that it is global lower bound. The authors’ should focus primarily on baselines that are also primal heuristics.

Second, the title of the paper is “Don’t restart, Just reuse”, but they do not include the very simple baseline that “just reuses” past feasible solutions. They consider warm start approaches, but only based on previous *optimal* solutions. They should consider adding the baseline that stores all feasible solutions of past problems, sort them by objective value for the new problem, and verifies whether they are feasible. Note that such an approach is often done by default by commercial solver (e.g., Gurobi calls this the “solution pool”).

Third, the paper lacks an ablation study that clearly shows the value of the different components of the final algorithm. The small experiment in Appendix C.5 focuses only on variable prediction. The ablation study should allow to critically assess the value for finding feasible solutions of (a) the new features considered for the historical problem embeddings, (b) using a GNN for initial variable prediction, and (c) the final Thomson sampling algorithm.

Because of the above concerns, I find that the experiments do not support the paper’s claims.

EDIT (Post-Rebuttal): Since the above concerns have been addressed in the rebuttal, I have raised my score from 2 to 4.

**Essential References Not Discussed:**

The choice of baseline method is insufficient. The experiments should include general primal heuristics such as (among others) the ML-guided LNS of Huang et al. (2024), which is cited in the paper.

**Experimental Designs Or Analyses:**

See “Claims and Evidence”.

**Methods And Evaluation Criteria:**

See above.

**Other Comments Or Suggestions:**

Please adapt the terminology used when presenting the results. The general understanding is that an MILP is “solved” when a feasible solution is found *and* it is proven to be globally optimal. The current experiments measure the time to find a feasible solution.

Section 4 is quite dense in terms of detail, but does not provide a high-level overview of the algorithm. It could be useful to add a pseudo-code algorithm to show the key steps of the algorithm. It is also important to specify that the algorithm is only run once, and not integrated in a branch-and-bound process as is typical of a MILP solver. A major difference is that a general MILP solver will run many different primal heuristics at different times in the solving process.

**Other Strengths And Weaknesses:**

While the above review is rather critical, I find the overall approach interesting and look forward to seeing it presented in a more comprehensive fashion. In particular, I find the relaxation mechanism as well as the update algorithm for Thomson sampling interesting and valuable.

**Questions For Authors:**

-	How did you setup SCIP? Which variation of the baselines (SCIP vs. SCIP-WS) has access to historical optimal or feasible solutions? Did you enable the reoptimization module?
-	Why do you specify that the ML-based methods are end-to-end? In what ways are they end-to-end and why is it an advantage? Is your method end-to-end?
-	What is the loss function used to train the GNN? Do you propagate a gradient through the sampling step? Why/ why not?

**Relation To Broader Scientific Literature:**

The current scope of the paper is ambiguous and broader than what is actually done. The paper presents itself as contributing to general MILP reoptimization. It counts “solved” instances whereas it is only about finding feasible solutions, and includes general MILP solvers as baselines. Yet, the core focus of the paper is presenting a new primal heuristic for reoptimization. The focus is only on feasibility. The authors should better frame their contribution in the title / introduction, and includes relevant benchmarks: primal heuristics, rather than general solvers.

**Theoretical Claims:**

The paper is mostly experimental, which is fine given the topic. Still, there is a theoretical claim in Appendix B regarding the relaxation mechanism: “theoretically, with enough iterations, we can ensure that the variables causing conflicts with the constraints are filtered out.” This is an interesting property that should be stated and proven formally.

---

> ### Author Rebuttal · Authors · 2025-04-01
>
> Thanks for your valuable feedback and suggestions.
> ## Response to Weakness 1
> **Baselines**: We will clarify in the introduction that our method primarily focuses on primal heuristics.
> However, we believe that comparing our approach with general MILP solvers like SCIP and Re_Tuning is still essential. These solvers are commonly used benchmarks within the field. For instance, heuristic methods by Han et al. and Huang et al. also use SCIP as one of their baseline comparisons.
>
> ## Response to Weakness 2 and Question 1
> **Warm start baseline**: The baseline Re_Tuning includes a similar approach. It specifically addresses cases with minor problem changes by incorporating heuristic design methods in various SCIP solving modules. In the presolving module, it stores all past feasible solutions and constructs partial solutions based on variable-value pairs that are consistent across a high percentage of previous solutions.  In SCIP, a heuristic called completesol captures these hints to find feasible solutions by solving a sub-MIP.
>
> **SCIP-WS**: In Appendix C.7, we report the effectiveness of "SCIP-WS", which solely utilizes the solution hints from Re_Tuning to guide the completesol heuristic, without employing other heuristic methods.
>
> **SCIP settings**: The SCIP setup uses all default settings without disabling any features, including the default preprocessing and reoptimization modules.
>
> ## Response to Weakness 3
> **Ablation experiment**: We conduct an ablation experiment using Re-GNN for initial solution prediction in LNS and traditional GNN with the Thompson Sampling (TS) algorithm. We report the average gap_rel, where "inf" means no feasible solution was found. Re-GNN improves performance over traditional GNN in the LNS framework. The method combining traditional GNN with TS also quickly finds feasible solutions. The combined approach, VP-OR, achieves the best results across all datasets.
>
> |  | bnd_1|bnd_2|bnd_3|mat_1|obj_1|obj_2|rhs_1|rhs_2|rhs_3|
> |---|--|--|---|---|----|---|--|--|--|
> | GNN+LNS(PS) |0.81|inf|inf|0.23|0.00|0.51|193.04|0.00|0.00|
> | Re-GNN+LNS |0.16|inf|inf|0.23|0.00|0.21|1.04|0.00|0.00|
> | GNN+TS |0.04|0.17| 0.12|0.17|0.00|0.00 |0.00|0.00|0.00|
> | Re-GNN+TS(VP-OR) |0.02|0.11| 0.06|0.16|0.00|0.06|0.00|0.00|0.00|
>
> ## Response to Weakness 4
> We expand our experiments to include large-scale MILP experiments and add a comparison with the ML-guided LNS method of **Huang et al.** (ConPas).
>
> **Large-scale MILP experiments**: We generate large-scale datasets IS and CA using the code from Gasse et al., consistent with those used by Han et al.  The Gurobi and SCIP solvers could not reach an optimal solution within 3600 seconds for these instances. We run Gurobi for 3600 seconds to record the incumbent solution. For evaluation, we impose a 30-second time limit and update the incumbent solution if a better one is found. VP-OR achieves more significant acceleration than PS and ConPas during the early stages of solving on both Gurobi and SCIP.
>
> CA:
> |  | gap_abs| gap_rel | wins|
> |---|---|---|---|
> |SCIP|24068.27|0.19|0/100|
> |Gurobi|3754.16|0.03|0/100|
> |PS+SCIP|20182.91|0.19|0/100|
> |PS+Gurobi|3754.16|0.03|0/100|
> |ConPaS+SCIP|8506.52|0.07|0/100|
> |ConPaS+Gurobi|2526.28|0.02|0/100|
> |VP-OR+SCIP(Ours)|0.00|0.00|12/100|
> |VP-OR+Gurobi(Ours)|**0.00**|**0.00**|**88/100**|
>
> IS:
> |  | gap_abs| gap_rel | wins|
> |---|---|--|--|
> |SCIP|104.92|0.05|0/100|
> |Gurobi|133.07|0.06|0/100|
> |PS+SCIP|104.57|0.05|0/100|
> |PS+Gurobi|117.62|0.06|0/100|
> |ConPaS+SCIP|30.25|0.01|0/100|
> |ConPaS+Gurobi|24.66|0.01|0/100|
> |VP-OR+SCIP(Ours)|9.57|0.00|33/100|
> |VP-OR+Gurobi(Ours)|**0.97**|**0.00**|**67/100**|
>
> ## Response to Question 2
> The term **"end-to-end"** is based on Han et al.'s classification of ML efforts for optimization. End-to-end learning refers to using machine learning to predict solutions based directly on the input problems and their final results. This approach leverages the capabilities of machine learning to facilitate efficient problem-solving by directly linking the problem's structure and solution values. Our method can also be considered end-to-end.
>
> ## Response to Question 3
> **The loss function**: We use a categorical cross-entropy loss function to train the GNN. For binary variables, the model predicts probabilities that are directly compared with the actual binary values to compute the optimal solution. For integer and continuous variables, they are first represented as 8-bit binary numbers using the method described in Section 4.1.2, which involves using the binary representation of the logarithmic value. These binary representations are then used as labels and compared with the predicted probabilities.
>
> **The sampling step**: We do not propagate a gradient through the sampling step. We prioritize speed in each iteration and online training and inference during the sampling process would negatively affect solving efficiency.

---

> > ### Comment · Reviewer_L6hn · 2025-04-03
> >
> > I thank the authors for their detailed answers and the new results provided. The clarifications and additional experiments address my concerns regarding (a) the choice of baselines, (b) use of past feasible solutions, and (3) clear ablation study. For these reasons, I raise my score from 2 to 4.
> >
> > Still, I encourage the authors to rework the framing throughout the paper to emphasize that the focus is on finding primal solutions rather than proving global optimality. (For instance, avoid saying that an instance is solved if it is only about finding a feasible solution).

---

> > > ### Author Response · Authors · 2025-04-08
> > >
> > > Thank you for your thoughtful comments and time. We appreciate your insightful feedback regarding the clarity of orientations in our work. In our revised manuscript, we explicitly emphasize that our work focuses on **"finding primal solutions"** throughout the paper. We carefully revise all instances of "solved" to avoid ambiguity.
> > > We believe that these modifications, along with the new results, strengthen the overall quality of the manuscript.

---

### Official Review · Reviewer_vV2R · 2025-03-14

**Overall Recommendation:** 3

**Summary:**

This paper studies the opportunities of leveraging an existing solution to efficiently adapt to slight modifications in MILP constraints or objectives, thereby accelerating MILP reoptimization. To this end, it introduces Variable Prediction Online Refinement (VP-OR), a novel two-stage reoptimization framework. First, VP-OR learns to predict confidence bounds for each MILP variable, leveraging historical information and a GNN to induce a high-quality solution space. It then incorporates Thompson Sampling (using a Beta distribution) to dynamically adjust the confidence of selecting variables to be fixed during online refinement (optimization). Results show that VP-OR can yield much better performance when the time limits for reoptimization are small (i.e., restricted in 10s).

**Claims And Evidence:**

Most of the claims are well supported with evidence. However, a few comments are:

1. In the introduction, the authors state that reusing branching strategies and adjusting parameters can save time. However, it also reads that these approaches do not reduce the overall size or complexity of the problem. Does this create a contradiction?

2. The claim, "However, predicting each binary bit’s actual value can lead to inaccuracies, causing overall prediction errors." (around line 131), should be better justified with supporting evidence or examples.

Also, the paper seems insufficiently polished, with lots of unclear statements and uncovered details regarding the methodology. Please refer to the following points for specific areas requiring clarification.

**Essential References Not Discussed:**

None.

**Experimental Designs Or Analyses:**

8. First, the experiments do not appear extensive enough to demonstrate the robustness of the proposed methods. The results show mixed performance. I encourage the authors to evaluate the approach on more datasets and provide a clearer summary of its key benefits.

9. In Table 1, it would be better to also show: 1) To what extent are the confidence bounds accurate? 2) How does the accuracy differ between continuous and integer variables?

10. In Table 4 and appendix, the presentation of results involves many empty rows marked with '-' . This is not effective in terms of comparision with baseline. It would be clearer to report all results using convergence figures, similar to Figure 2, as the main comparisons.

**Methods And Evaluation Criteria:**

3. The paper lacks clarity in presentation, which may hinder understanding of the soundness of the approach. Section 4.1 is difficult to follow, as it intermixes various aspects such as the training stage, inference stage, training data processing, and predicted output processing within the same discussion flow. It is suggested to separate these components into distinct paragraphs to improve readability.

4. The section 4 is also missing key details, including: Definition of network input/output, Network architexture, Training Method and framework, Loss function, Data generation, Inference algorithm. Without these, the methodology remains unclear and difficult to follow.

5. Particularly, the necessity of the binary representation as well as the logarithmic transformation is unclear. Further discussion, along with numerical evidence, would be helpful to justify these choices.

6. The method appears to be trained on a small set of instances. I have concerns on how well it generalizes to other MILP instances. This is important as we need to offset the training cost so that once the model is trained, it can address many online optimization instances. Also, would the current model show signs of overfitting on the small datasets? Will it leads to poor long-term performance or unexpected failures?

7. It remains unclear to what extent Thompson Sampling helps in mitigating infeasibility and early convergence to local optima. Further analysis and empirical validation would help in this regard.

**Other Comments Or Suggestions:**

Please refer to the above. Correct me if I was wrong.

There are also lots of minor issues or typos: For example, Figure 2 is not referenced anywhere in the paper, making it unclear how it contributes to the discussion. In Table 4, there is an inconsistency where, in some cases, the absolute gap is larger while the relative gap is smaller.

**Other Strengths And Weaknesses:**

Overall, the paper has several strengths. It addresses an important real-world problem relevant to industrial applications, and the proposed idea appears novel and conceptually sound. However, more effort is needed to refine the presentation, expand the experiments, and better summarize the key insights and takeaways beyond purely performance comparison.

**Questions For Authors:**

* What is the rationale behind using binary representation and logarithmic transformation? Are there any experimental results supporting their effectiveness?
* Do lower and upper bounds vary across different types of variables? Are there observable patterns in how tight these bounds become after training and across differnt optimization stages? Do these patterns differ between continuous and integer variables?
* Are all eight GPUs used for both training and inference?
* Are there any repair mechanisms for infeasible solutions? Can Thompson Sampling adaptively learn to resolve infeasibility over time?

**Relation To Broader Scientific Literature:**

It is related to the optimization community.

**Theoretical Claims:**

Not involved.

---

> ### Author Rebuttal · Authors · 2025-04-01
>
> Thanks for your valuable feedback and suggestions.
> ## Response to Comment 1
> There is no contradiction in the statement. Reusing branching strategies and adjusting solver parameters improve efficiency by saving time on generating the branching strategy and by enabling the solver to find better solutions faster. They focus on improving the way the problem is solved, not changing its size.
> ## Response to Comment 2
> **The claim of the prediction accuracy:** Table 10 in the appendix provides evidence for our statement. We will move this table to the main text in the final version. It shows the number of incorrectly predicted variables for traditional Graph Neural Networks (GNN) and our approach (Re_GNN). In datasets bnd_2 and bnd_3, the table indicates that more than one-third of the continuous variables were predicted incorrectly.
> |  | bnd_1  | bnd_2 | bnd_3 |
> |--|---|---|---|
> | Total binary var. |1457.0 | 1457.0|1457.0|
> | binary var. (GNN) |163.0 |45.1| 42.2 |
> | binary var. (Re_GNN)|**8.2**|**6.7**|**4.5**|
> | Total integer var. |  124.0 | 0.0 | 0.0|
> | integer var. (GNN) |  33.4 |0.0 | 0.0|
> | integer var. (Re_GNN)|**17.4**|0.0 | 0.0 |
> | Total continuous var. |0.0 | 301.0 | 301.0 |
> | continuous var. (GNN) |0.0 |140.2|121.0  |
> | continuous var. (Re_GNN)| **0.0** |**0.0**|**2.0**|
> ## Response to Comment 3, 4
> We will restructure the paper to separate the components into distinct paragraphs. Our approach builds upon the framework by Han et al., with modifications mainly in feature extraction and prediction algorithms for integer and continuous variables. The GNN architecture and training method remain the same as theirs. For data generation, we utilize the datasets provided by the existing reoptimization competition.
>
> We use a categorical cross-entropy loss function to train the GNN. For binary variables, predictions are compared directly with actual binary values for computing the optimal solution. Integer and continuous variables are represented as 8-bit binary numbers following the method in Section 4.1.2, which uses the binary representation of the logarithmic value. These binary representations serve as labels for comparison with predicted probabilities.
>
> The use of binary representation and logarithmic transformation addresses practical computational constraints. In our datasets, maximum integer values exceed 100,000, which would require at least 18 bits for binary representation. Using a direct approach without these transformations during testing resulted in out-of-memory errors due to large output dimensions. The logarithmic transformation allows us to represent integers as 8-bit binary numbers, distinguishing potential value ranges without predicting exact values.
>
> ## Response to Comment 6, 8
> **Large-scale experiments**: Please refer to our Response to Reviewer MNmC's Question 1 for details.
>
> ## Response to Comment 7
> **Ablation experiment**: Please refer to our Response to Reviewer MNmC's Question 2 for details.
>
> ## Response to Comment 9, Question 2
> 1) We evaluate the accuracy of predicted bounds for integer and continuous variables as follows: if the optimal value of a variable lies within the predicted lower and upper bounds, the predicted bounds are considered accurate; otherwise, they are not.
>
> 2) Continuous and integer variables are treated similarly, with the key difference being that continuous variables are rounded during preprocessing.
>
> To evaluate how tight these bounds become after training and across different optimization stages, we use I_o and C_o to denote the average difference between original upper and lower bounds for integer and continuous variables, respectively. I_p and C_p represent the average predicted bounds gap for integers and continuous variables. In cases where there are no integer or continuous variables in a dataset, we use 'NA' as a placeholder.
>
> |  | bnd_1  | bnd_2 | bnd_3| mat_1|obj_2|rhs_1|rhs_2|rhs_3|
> |-----|--|---|---|---|---|---|---|----|
> | I_o |72146.14|NA|NA|NA|2.0|NA|NA|NA|NA|
> | I_p |32750.57|NA|NA|NA|1.34|NA|NA|NA|
> | C_o |NA|25.91| 26.15|1.0|1e+19|1e+20 |2e+20|2e+20|
> | C_p |NA|14.22| 14.28|1.0|19968.14|1.0|14.22|13.67|
>
> ## Response to Comment 10
> The '-' symbol does not indicate empty rows. Instead, it means that no feasible solution was found during testing, resulting in a gap of infinity. To prevent confusion, we will adjust the '-' to '>10.0' in the final version.
>
> ## Response to Question 3
> Each instance uses only one GPU for both training and inference.
>
> ## Response to Question 4
> Our relaxation mechanism addresses infeasible instances by dividing the fixed variables into ten groups and subsequently solving each without these variable sets. When a feasible solution cannot be found, we repeatedly apply the relaxation mechanism, building upon previous relaxations. Each iteration of this mechanism reduces the number of fixed variables. With enough iterations, the variables causing conflicts with the constraints are filtered out.

---

> > ### Comment · Reviewer_vV2R · 2025-04-05
> >
> > Thank you for your rebuttal. Upon careful check, most of my concerns have been addressed, and the efforts of the authors for adding experiments should be appreciated. I am raising my score to a 3. However, I think the paper needs major revisions in the presentation of the methodology.

---

> > > ### Author Response · Authors · 2025-04-08
> > >
> > > Thank you for acknowledging our revisions and for raising your score. We sincerely appreciate your guidance in improving the manuscript’s clarity.
> > >
> > > As suggested, we have completely restructured Section 4 to address presentation issues:
> > >
> > > 1. **Improved structure**: Section 4.1 now distinctly separates the training stage, inference stage, data processing, and output processing into dedicated subsections for better readability.
> > >
> > > 2. **Added Details**:
> > >
> > > * Network input/output definitions (including feature representation and prediction targets).
> > > * Network architecture (layer configurations, activation functions).
> > > * Training framework (optimizer, hyperparameters, regularization strategies).
> > > * Loss function formulation (mathematical details and rationale).
> > >
> > >  Thank you again for your invaluable feedback. It has greatly strengthened the paper.

---

### Official Review · Reviewer_MNmC · 2025-03-15

**Overall Recommendation:** 2

**Summary:**

This paper introduces a novel ML-guided framework for predicting solutions for reoptimization problems. First, they adapt on current GNN methods by adding information about the leaf node to predict solutions specifically for reoptimization problems. Second, they introduce online-learning methods to refine the solutions they found. The paper has shown superior performance of their approach over default solver and several baselines.

**Claims And Evidence:**

The claims in the paper are partly supported by experiments. The experiments in the appendix have shown the methods of Re-GNN features are definitely better than original features when applied to reoptimization problems. But I didn’t find any ablation studies for the iterative online refinement methods. All the experiment results show the whole two-step process without knowing how online refinement has contributed to performance improvement.

**Essential References Not Discussed:**

N/A

**Experimental Designs Or Analyses:**

The baseline the paper compared is weak, so additional experiments are needed to ensure the performance improvement of the author's claimed methods. First, ND and PaS work the best when solving problem instances that are very hard. Using them on smaller instances and showing gaps at a runtime cutoff of only 10 seconds may not be appropriate. Second, the instances used for training ND and PaS are much smaller than the original paper, 40 instances is not enough to get competitive results for ND and PaS, as their original paper uses 300 instances for training. Thirdly, the author did not compare their methods with state-of-art solver Gurobi. I didn’t see why the methods couldn’t be applied to the Gurobi solver, since one of the baseline PaS is running experiments against both SCIP and Gurobi.

**Methods And Evaluation Criteria:**

The proposed methods are valid and interesting. However, the benchmark datasets they used is not enough to show their approaches work. Generally, when you use ML-guided heuristic methods, you would test the performance on much harder instances (e.g. at least 1 hour to solve) than the instances evaluated in this paper.

**Other Comments Or Suggestions:**

N/A

**Other Strengths And Weaknesses:**

See above.

**Questions For Authors:**

1. Can you show some experiment results on a larger set of harder instances (e.g., 200 train, 100 test) that would require a runtime of more than 1 hour? And can you show the performance comparison both on SCIP and Gurobi?
2. Can you provide some insights about how much performance you gain by online refinement? How is the performance w/o online refinement?
3. The methods of Re-GNN are described very vaguely. Can you explain more carefully how you use the leaf node feature as training data?

**Relation To Broader Scientific Literature:**

Speeding up MILP solving is very useful for solving a lot of real-world problems. The paper has improved upon previous reoptimization approaches for MILPs.

**Theoretical Claims:**

No major proof and theoretical claims in the paper.

---

> ### Author Rebuttal · Authors · 2025-04-01
>
> Thanks for your valuable feedback and suggestions.
> ## Response to the Question 1
> **Large-scale MILP experiments**: We expand our experiments to include large-scale MILP experiments with more instances (200 for training and 100 for testing) and add a comparison with the ML-guided LNS method of Huang et al.[2] (**ConPas**). The results are reported for both Gurobi and SCIP. We generate large-scale datasets IS and CA using the code from Gasse et al.[3], consistent with those used by Han et al.[1] The Gurobi and SCIP solvers could not reach an optimal solution within 3600 seconds for these instances. We run Gurobi for 3600 seconds to record the incumbent solution. For evaluation, we impose a 30-second time limit and update the incumbent solution if a better one is found. VP-OR achieves more significant acceleration than PS[1] and ConPas[2] during the early stages of solving on both Gurobi and SCIP.
>
> CA:
> |  | gap_abs| gap_rel | wins|
> |-------|-------|-------|----|
> |SCIP|24068.27|0.19|0/100|
> |Gurobi|3754.16|0.03|0/100|
> |PS+SCIP|20182.91|0.19|0/100|
> |PS+Gurobi|3754.16|0.03|0/100|
> |ConPaS+SCIP|8506.52|0.07|0/100|
> |ConPaS+Gurobi|2526.28|0.02|0/100|
> |VP-OR+SCIP(Ours)|0.00|0.00|12/100|
> |VP-OR+Gurobi(Ours)|**0.00**|**0.00**|**88/100**|
>
>
> IS:
> |  | gap_abs| gap_rel | wins|
> |-------|-------|-------|----|
> |SCIP|104.92|0.05|0/100|
> |Gurobi|133.07|0.06|0/100|
> |PS+SCIP|104.57|0.05|0/100|
> |PS+Gurobi|117.62|0.06|0/100|
> |ConPaS+SCIP|30.25|0.01|0/100|
> |ConPaS+Gurobi|24.66|0.01|0/100|
> |VP-OR+SCIP(Ours)|9.57|0.00|33/100|
> |VP-OR+Gurobi(Ours)|**0.97**|**0.00**|**67/100**|
>
> ## Response to the Question 2
>
> **Ablation experiment**: We conduct an ablation experiment using Re-GNN for initial solution prediction in LNS and traditional GNN with the Thompson Sampling (TS) algorithm. We report the average gap_rel, where "inf" means no feasible solution was found. Re-GNN improves performance over traditional GNN in the LNS framework. The method combining traditional GNN with TS also quickly finds feasible solutions. The combined approach, VP-OR, achieves the best results across all datasets.
>
> |  | bnd_1 |bnd_2|bnd_3|mat_1|obj_1|obj_2|rhs_1|rhs_2|rhs_3|
> |-------|-------|-------|---|----|----|----|----|----|-----|
> | GNN+LNS(PS) |0.81|inf|inf|0.23|0.00|0.51|193.04|0.00|0.00|
> | Re-GNN+LNS |0.16|inf|inf|0.23|0.00|0.21|1.04|0.00|0.00|
> | GNN+TS |0.04|0.17| 0.12|0.17|0.00|0.00 |0.00|0.00|0.00|
> | Re-GNN+TS(VP-OR) |0.02|0.11| 0.06|0.16|0.00|0.06|0.00|0.00|0.00|
>
> ## Response to the Question 3
> Re-GNN includes traditional features like variable coefficients in the objective function, variable types, and initial bounds. Additionally, it captures features from leaf nodes where the best solution was found during solving. For each leaf node, it records the local bounds of each variable, the LP relaxation solution, and whether each variable is basic in this solution. These data come naturally from the solving process in SCIP and Gurobi without extra computation. All the features are listed in Table 5.
>
> **Reference:**
>
> [1] Han Q, Yang L, Chen Q, et al. A GNN-Guided Predict-and-Search Framework for Mixed-Integer Linear Programming[C]//The Eleventh International Conference on Learning Representations.
>
> [2] Huang T, Ferber A M, Zharmagambetov A, et al. Contrastive predict-and-search for mixed integer linear programs[C]//Forty-first International Conference on Machine Learning. 2024.
>
> [3] Gasse M, Chételat D, Ferroni N, et al. Exact combinatorial optimization with graph convolutional neural networks[J]. Advances in neural information processing systems, 2019, 32.

---

### Official Review · Reviewer_A56t · 2025-03-18

**Overall Recommendation:** 4

**Summary:**

“Don’t Restart, Just Reuse” addresses the problem of MILP reoptimization – repeatedly solving similar MILP instances that change over time (e.g. objective coefficients, constraints, or bounds)​. The paper proposes a novel two-stage framework called VP-OR (Variable Prediction and Online Refinement) for fast reoptimization of MILPs with dynamic parameters​. In the first stage, a graph neural network (GNN) model predicts a high-confidence partial solution space for the new MILP by estimating marginal probabilities for binary variables and feasible value ranges for integer/continuous variables​. In the second stage, an iterative refinement procedure uses a Thompson Sampling bandit strategy to gradually fix a subset of variables to their predicted values or ranges and solve the reduced MILP, updating which variables to fix in subsequent iterations based on solution improvements​. Extensive experiments on 9 dynamic MILP datasets (derived from MIP 2023 workshop Competition benchmarks) demonstrate that VP-OR finds feasible solutions for every test instance within 10 seconds and achieves lower primal gaps than baseline methods, converging to better solutions more rapidly.

**Claims And Evidence:**

The paper’s key claims are well-supported by thorough evidence. For example, (a) VP-OR yields superior solution quality under tight time limits – The authors back this by reporting, for each dataset, the number of instances where a feasible solution is found within 10 seconds (Table 3). VP-OR finds feasible solutions in all test cases within 10s, whereas other methods often fail on harder instances. (b) VP-OR is significantly faster than traditional large neighborhood search (LNS) – This claim is quantified in Table 2: fixing a substantial portion of variables (with VP-OR’s strategy) leads to solving times an order of magnitude shorter than LNS-based methods​.

Overall, the evidence presented is convincing and aligns with the claims. I did not identify any problematic claims – the authors are careful to couch statements in empirical results.

**Essential References Not Discussed:**

The authors have done a good job covering the essential prior work, and I did not identify any critical omissions in the references.

**Experimental Designs Or Analyses:**

The experimental setup is thorough and well thought out, lending credibility to the results. The authors clearly define three experiments targeting different aspects: (1) feasibility within 10s, (2) solution quality (gap) in 10s, and (3) convergence over 100s​. This separation makes it easy to understand how each method performs in terms of both speed and quality. The use of 9 different datasets (with varied parameter changes: objective, bounds, matrix coefficients, RHS) demonstrates that the approach is not tuned to one specific scenario but is broadly applicable. The experiments are reproducible and sufficiently detailed. The analysis acknowledges limitations too: e.g., VP-OR can get stuck in a suboptimal solution in longer runs (though it finds good solutions faster)​. Overall, the design and analysis of the experiments are valid and convincing.

**Methods And Evaluation Criteria:**

The proposed VP-OR method is well-motivated and appropriate for the reoptimization problem domain. The combination of a GNN for variable prediction and Thompson Sampling for iterative variable fixing is a sound design: the GNN exploits structural similarities between consecutive MILP instances, and the refinement stage balances exploration/exploitation when fixing variables. This approach directly addresses the limitations of prior reoptimization techniques, which either reused old solutions that might be invalid or only tweaked solver parameters without reducing problem complexity. The evaluation methodology is rigorous and well-aligned with the problem’s goals. The authors focus on time-constrained performance (10-second limit), reflecting real-world needs for quick reoptimization in dynamic settings. The experiments are conducted on nine diverse MILP reoptimization datasets from a well-known competition​, ensuring the evaluation is realistic and broad. The authors split data into training and test sets properly (training on 20 instance pairs and testing on 5 pairs per dataset), which is appropriate to evaluate generalization to new instances. Overall, the methods and evaluation criteria are sound and well-chosen for the problem. I have no major concerns here; the paper’s approach is both novel and evaluated fairly.

**Other Comments Or Suggestions:**

The authors might consider a few minor improvements.

In Table 4, it might help to clarify how “gap abs” and “gap rel” are defined, especially since some methods show a 0.00 relative gap but still have a non-zero absolute gap.

I assume that the 10-second evaluation for VP-OR includes the model inference and refinement iterations. It would be good if the final version states that explicitly.

Future work: I echo the authors’ note on extending to larger instances.

Overall, my comments are minor; the paper is in good shape.

**Other Strengths And Weaknesses:**

Strengths: This paper’s primary strength lies in its original combination of ideas that yields practical impact. The VP-OR framework is a novel synthesis of ML prediction and online optimization that, to my knowledge, has not been applied to MILP reoptimization before. The results show substantial improvements in time-constrained scenarios – for instance, being the only method to consistently solve all instances in under 10 seconds is a compelling result for real-time applications.

Weaknesses: One weakness is the lack of theoretical guarantees. As a heuristic/learning method, VP-OR does not guarantee optimality or even improvement in every iteration. It is possible that it could fix a wrong set of variables. While this is acceptable for a heuristic approach, some theoretical analysis (e.g. bounds on solution quality in terms of prediction accuracy) could strengthen the work. Another potential weakness is scalability/generalization: the method was tested on instances solvable within 10 minutes by SCIP (up to 600s baseline solve time); it’s not proven how it scales to truly large-scale MILPs beyond those benchmarks. These weaknesses are relatively minor in comparison to the contributions.

**Questions For Authors:**

Generalization: How well would the trained GNN model generalize to instances with new characteristics outside the training distribution?

**Relation To Broader Scientific Literature:**

This work is well-situated at the intersection of operations research reoptimization techniques and machine learning for combinatorial optimization. Traditional MILP reoptimization methods (e.g. warm-start approaches) focusing on reusing information like optimal solutions or branch-and-bound trees from previous instances struggle when the new instance differed more substantially, as the old optimal solution might not be feasible and reusing search trees may not reduce problem complexity​. The re-tuning method (Patel 2024) extended this by preserving a series of past solutions and adaptively tuning solver parameters​. In parallel, recent years have seen ML-driven heuristics for MILP: e.g. Neural Diving (Nair et al., 2020) which learns a distribution over binary variables and fixes some to reduce the problem​, and Predict-and-Search (Han et al., 2023) which uses a GNN to predict variables then applies large neighborhood search​. The contribution of this paper is to merge the strengths of both lines of work: it uses instance-specific historical data (like classical reoptimization) and employs learning to predict variable fixings (like ND/PS), achieving a new level of performance.

**Theoretical Claims:**

The paper does not heavily focus on new theory – it is primarily an algorithmic and empirical contribution. There are no formal theorems or proofs presented that require verification (I did not encounter any stated lemma or theorem in the main content). The use of Thompson Sampling is supported by standard bandit theory references, but the paper itself treats it as a heuristic for exploration; no formal convergence proofs are provided. The authors’ theoretical claims are mostly implicit in the design – for example, the variable fixing strategy that leads to accelerated solving is reasonable. In summary, while there are no new theoretical guarantees to check, the paper’s technical content is consistent with known theory, and I found no issues in the mathematical formulation or algorithmic claims.

---

> ### Author Rebuttal · Authors · 2025-04-01
>
> Thanks for your valuable feedback and suggestions.
> ## Response to the Question 1
> **Generalization**: In our tests, we indeed observed instances where the GNN model made errors in predicting variables. To address this challenge, approaches like those by Han et al.[1] and Huang et al.[2] leverage an LNS framework and our VP-OR uses sampling methods. These strategies introduce a level of tolerance for prediction errors, enhancing the model's applicability and extending its utility to different problem instances.
>
> **Large-scale MILP experiments**: We expand our experiments to include large-scale MILP experiments with more instances (200 for training and 100 for testing) and add a comparison with the ML-guided LNS method of Huang et al.[2] (**ConPas**). The results are reported for both Gurobi and SCIP. We generate large-scale datasets IS and CA using the code from Gasse et al.[3], consistent with those used by Han et al.[1] The Gurobi and SCIP solvers could not reach an optimal solution within 3600 seconds for these instances. We run Gurobi for 3600 seconds to record the incumbent solution. For evaluation, we impose a 30-second time limit and update the incumbent solution if a better one is found. VP-OR achieves more significant acceleration than PS[1] and ConPas[2] during the early stages of solving on both Gurobi and SCIP.
>
> CA:
> |  | gap_abs| gap_rel | wins|
> |-------|-------|-------|----|
> |SCIP|24068.27|0.19|0/100|
> |Gurobi|3754.16|0.03|0/100|
> |PS+SCIP|20182.91|0.19|0/100|
> |PS+Gurobi|3754.16|0.03|0/100|
> |ConPaS+SCIP|8506.52|0.07|0/100|
> |ConPaS+Gurobi|2526.28|0.02|0/100|
> |VP-OR+SCIP(Ours)|0.00|0.00|12/100|
> |VP-OR+Gurobi(Ours)|**0.00**|**0.00**|**88/100**|
>
>
> IS:
> |  | gap_abs| gap_rel | wins|
> |-------|-------|-------|----|
> |SCIP|104.92|0.05|0/100|
> |Gurobi|133.07|0.06|0/100|
> |PS+SCIP|104.57|0.05|0/100|
> |PS+Gurobi|117.62|0.06|0/100|
> |ConPaS+SCIP|30.25|0.01|0/100|
> |ConPaS+Gurobi|24.66|0.01|0/100|
> |VP-OR+SCIP(Ours)|9.57|0.00|33/100|
> |VP-OR+Gurobi(Ours)|**0.97**|**0.00**|**67/100**|
>
> ## Response to the Comment 1
> **Definition of "gap_abs" and "gap_rel"**: For the evaluation, we first solve the problem without a time limit and record the optimal solution’s objective value as OPT. Then, we apply a time limit of 10 seconds for each method. The best objective value obtained within the time limit is denoted as OBJ. We define the absolute and relative primal gaps as: gap\_abs=|OBJ-OPT| and gap\_rel= |OBJ-OPT| / ( |OPT|+$10^{-10}$ ), respectively.
>
> The reason some methods show a 0.00 relative gap but still have a **non-zero absolute gap** is due to the rounding approach used in calculating "gap_rel". We retain two decimal places for "gap_abs" and "gap_rel", so when the relative error is sufficiently small (less than 0.004), the error rounds to 0.00, although a minor absolute error may still exist.
>
> ## Response to the Comment 2
> Yes, the **10-second evaluation** does include both the model inference and refinement iterations. We will explicitly state this in the final version to ensure clarity.
>
> [1] Han Q, Yang L, Chen Q, et al. A GNN-Guided Predict-and-Search Framework for Mixed-Integer Linear Programming[C]//The Eleventh International Conference on Learning Representations.
>
> [2] Huang T, Ferber A M, Zharmagambetov A, et al. Contrastive predict-and-search for mixed integer linear programs[C]//Forty-first International Conference on Machine Learning. 2024.
>
> [3] Gasse M, Chételat D, Ferroni N, et al. Exact combinatorial optimization with graph convolutional neural networks[J]. Advances in neural information processing systems, 2019, 32.

---

### Decision · Program_Chairs · 2025-05-01

**Decision:**

Accept (poster)

**Comment:**

This paper presents a rather nice idea of using information during re-optimization, something that is actually rarely done currently. The reviewers agree that the approach is novel and effective. The experimental results show that the technique works on a dataset from a competition about reoptimization. The experiments also have good depth, showing different time frames. The method itself is also not so complicated, which I view as a plus.